

# Global 30-m seamless data cube (2000-2022) of land surface reflectance generated from Landsat-5,7,8,9 and MODIS Terra constellations

Shuang Chen[1], Jie Wang[2], Qiang Liu[2], Xiangan Liang[3], Rui Liu[1], Peng Qin[4], Jincheng Yuan[2], Junbo Wei[2], Shuai Yuan[1], Huabing Huang[2,4], Peng Gong[1,5,6]

[1]Department of Geography, The University of Hong Kong, Hong Kong, China
[2]Peng Cheng Laboratory, Shenzhen 518000, China
[3]Ministry of Education Key Laboratory for Earth System Modeling, Department of Earth System Science, Tsinghua University, Beijing 100084, China
[4]School of Geospatial Engineering and Science, Sun Yat-sen University, Guangzhou 510275, China
[5]Department of Earth Sciences, The University of Hong Kong, Hong Kong, China
[6]Institute for Climate and Carbon Neutrality, The University of Hong Kong, Hong Kong, China

*Correspondence to*: Peng Gong (penggong@hku.hk) and Jie Wang (wangj10@pcl.ac.cn)

**Abstract.** The Landsat series constitutes an unparalleled repository of multi-decadal Earth observations, serving as a cornerstone in global environmental monitoring. However, the inconsistent coverage of Landsat data due to its long revisit intervals and frequent cloud cover poses significant challenges to land monitoring over large geographical extents. In this study, we developed a full-chain processing framework for the multi-sensor data fusion of Landsat-5, 7, 8, 9 and MODIS Terra surface reflectance products. Based on this framework, a global, 30-m resolution, and daily Seamless Data Cube (SDC) of land surface reflectance was generated, spanning from 2000 to 2022. A thorough evaluation of the SDC was undertaken using a leave-one-out approach and a cross-comparison with NASA's Harmonized Landsat and Sentinel-2 (HLS) products. The leave-one-out validation at 425 global test sites assessed the agreement between the SDC with actual Landsat surface reflectance values (not used as input), revealing an overall Mean Absolute Error (MAE) of 0.014 (the valid range of surface reflectance values is 0-1). The cross-comparison with the HLS products at 22 Military Grid Reference System (MGRS) tiles revealed an overall Mean Absolute Deviation (MAD) of 0.017 with L30 (Landsat-8-based 30-m HLS product) and a MAD of 0.021 with S30 (Sentinel-2-based 30-m HLS product). Moreover, experimental results underscore the advantages of employing the SDC for global land cover classification, achieving a sizable improvement in overall accuracy (2.4%~11.3%) over that obtained using Landsat composite and interpolated datasets. A web-based interface has been developed for researchers to freely access the SDC dataset, which is available at https://doi.org/10.12436/SDC30.26.20240506 (Chen et al., 2024).





## 1 Introduction

Earth Observation (EO) data acquired by satellite sensors are fundamental to global land monitoring (Markham and Helder, 2012; Song et al., 2018; Wulder et al., 2022), providing critical information sources with unparalleled spatial and temporal coverage at a low cost. Over the past decades, satellite remote sensing has emerged as a prominent technology in Earth system science (Gong et al., 2023; Yang et al., 2013), contributing to the monitoring of land surface dynamics (Gong et al., 2013, 2019; Huang et al., 2017; H. Liu et al., 2020; Liu et al., 2021; Song et al., 2018), land surface phenology (Bolton et al., 2020; Piao et al., 2019), forest (Hansen et al., 2013, 2008), water (Ji et al., 2018; Pekel et al., 2016; Pickens et al., 2022; Sagan et al., 2020), and urbanization (Gong et al., 2020, 2012; X. Liu et al., 2020).

The Landsat series stands as the most enduring source of Earth observations, with a historical archive extending back to 1972 (Wulder et al., 2022). This longevity, combined with its relatively high spatial resolution, rigorous radiometric calibration, and free-access policy, has made Landsat a cornerstone for monitoring global terrestrial environments (Markham and Helder, 2012; Wulder et al., 2022). Nevertheless, the utility of Landsat data inevitably encounters certain limitations. A notable constraint is its relatively low temporal frequency, revisiting each area on Earth every 16 days (8 days when there are two Landsat satellites in orbit with an 8-day offset) (Zhu et al., 2015). This issue is further compounded by the presence of cloud and cloud shadow, which can introduce significant temporal gaps in the acquisition of clear-sky observations, especially in cloudy regions (Zhu et al., 2016). Moreover, Landsat time series observations typically exhibit irregularities in both observation frequencies and acquisition dates, due to the presence of cloud contamination and the geographically heterogeneous Landsat overpass coverage (Li and Roy, 2017). These irregularities present significant challenges when utilizing Landsat for large-scale monitoring of land cover and land use change (Potapov et al., 2020; Zhang et al., 2024). Therefore, the availability of Landsat datasets characterized by consistency in both temporal and spatial dimensions is crucial for facilitating various global environmental studies (Khan et al., 2024; Li et al., 2023; Pickens et al., 2022; Potapov et al., 2022b, 2022a, 2021b, 2021a; Song et al., 2021; Turubanova et al., 2023)

One conventional approach employed to mitigate data gaps in optical remote sensing is image compositing, which selects the highest-quality observations within a pre-defined time interval, based on specific criteria, to create seamless clear images at large scales (Jin et al., 2023; Qiu et al., 2023; White et al., 2014). Historically, image compositing has been mostly applied to coarse resolution data with high temporal frequency (Qiu et al., 2023), such as that obtained by the Advanced Very High Resolution Radiometer (AVHRR) and the Moderate Resolution Imaging Spectroradiometer (MODIS) sensors (Chuvieco et al., 2005; Cihlar et al., 1994; Holben, 1986; Huete et al., 2002; Wolfe et al., 1998). The use of image compositing for medium resolution data (e.g. Landsat) was comparatively uncommon before the advent of Landsat free-access policy in 2008 (Qiu et al., 2023). In recent years, many image compositing algorithms have been developed for Landsat data (Frantz et al., 2017; Griffiths et al., 2019; Jin et al., 2023; Nelson and Steinwand, 2015; Qiu et al., 2023; Roy et al., 2010; White et al., 2014). Nevertheless, Landsat image compositing is not without its limitations. Firstly, due to the lack of frequent Landsat observations (especially in cloudy areas), it may take several months or even years to provide a



composite Landsat image, which can cause problems if there are land cover or phenological changes (Zhu et al., 2015). Furthermore, the compositing process may introduce distortions to the temporal dynamics of Landsat time series (Qiu et al., 2023), thereby hampering subsequent applications that depend on precise temporal information.

Landsat interpolation methods also provide the capability to generate seamless synthetic Landsat images using some prior-based interpolation models (Brooks et al., 2012; Malambo and Heatwole, 2016; Yan and Roy, 2018, 2020; Zhu et al., 2015). Despite their simplicity in comprehension and implementation, a significant limitation of these interpolation-based methods is their dependence on numerous clear-sky Landsat observations for accurate time series estimation (Chen et al., 2021; Zhu et al., 2015). This requirement poses a considerable obstacle to their large-scale applications, particularly in cloudy areas. Moreover, the performance of interpolation-based methods relies on the careful tuning and selection of model parameters, thereby encountering the challenge of balancing between over-fitting and under-fitting (Wu et al., 2022; Zhou et al., 2022). Large-scale remote sensing applications prefer processing algorithms capable of automatic adaptation to diverse input data conditions, eliminating the need for manual parameter tuning (Chen et al., 2023).

The spatiotemporal fusion technique provides a promising solution, which aims at incorporating more frequent coarse-resolution observations to enhance the temporal frequency of Landsat and generate synthetic Landsat-like dense time-series images (Chen et al., 2023, 2021; Gao et al., 2006; Liu et al., 2022; Zhu et al., 2016, 2010). For example, the Terra/Aqua Moderate Resolution Imaging Spectroradiometer (MODIS) provides frequent coarse-resolution observations at a 250-500-1000 m spatial resolution with a near-daily revisit frequency (Schaaf et al., 2002). The MODIS land bands have comparable center wavelengths to the Landsat Enhanced Thematic Mapper Plus (ETM+) sensor, making the MODIS data an ideal input for the spatiotemporal fusion with Landsat (Gao et al., 2006). Many different types of Landsat-MODIS spatiotemporal fusion methods have been developed (Chen et al., 2023; Gao et al., 2006, 2022; Goyena et al., 2023; Guo et al., 2020; Hilker et al., 2009a; Liu et al., 2019, 2022; Mizuochi et al., 2017; Shi et al., 2022; Wang et al., 2017, 2020; Zhang et al., 2013; Zhu et al., 2010, 2016; Zurita-Milla et al., 2008) and applied to land cover and land surface phenology monitoring (Abowarda et al., 2021; Battude et al., 2016; Chen et al., 2018; Gervais et al., 2017; Hilker et al., 2009b; Y. Li et al., 2017; Senf et al., 2015; Singh, 2011; Tian et al., 2013; Walker et al., 2012; Watts et al., 2011). The utility of multi-sensor data fusion in facilitating land cover and land use analyses has also been validated empirically in previous studies (Carrasco et al., 2019; Chen et al., 2017; Yin et al., 2019).

In this study, we: (i) developed a full-chain processing framework for the multi-sensor data fusion of Landsat-5, 7, 8, 9 and MODIS Terra surface reflectance products; (ii) generated a global, 30-m, and daily Seamless Data Cube (SDC) of land surface reflectance, covering the period from 2000 to 2022; (iii) evaluated the reconstruction accuracy of the proposed framework quantitatively using a leave-one-out strategy at 425 global test sites; (iv) evaluated the quality of the SDC quantitatively by cross-comparing it with the Harmonized Landsat and Sentinel-2 (HLS) products at 22 Military Grid Reference System (MGRS) tiles; (v) evaluated the performance of using the SDC for global-scale land cover classification against Landsat composite and interpolated datasets; (vi) provided a web-based interface for researchers to freely access the SDC dataset.



## 2 Materials

### 2.1 Landsat collection-2 level-2 surface reflectance products

We collected a comprehensive dataset comprising 6,564,546 Landsat Collection 2 Level 2 Surface Reflectance (L2SR) images from the USGS Earth Resources Observation and Science (EROS) Center, including data acquired by the Landsat 8-9 Operational Land Imager (OLI), Landsat 7 Enhanced Thematic Mapper Plus (ETM+), and Landsat 5 Thematic Mapper (TM). This dataset covers most of global land surface except Antarctica, spanning from 2000 to 2022.

The L2SR products are generated through a sequence of processing steps applied to Landsat raw data. These steps include reprojection, radiometric calibration, geometric correction, atmospheric correction, and cloud masking. Compared to the previous Landsat Collection 1 products, the Collection 2 products have markedly improved the Landsat absolute geolocation accuracy using Landsat 8 geolocational imaging performance harmonized with the ESA GRI data (Crawford, 2023). The Landsat 5,7 TM/ETM+ data were atmospherically corrected using the Landsat Ecosystem Disturbance Adaptive Processing System (LEDAPS) (Masek et al., 2006), and the Landsat 8-9 OLI data were corrected using the Land Surface Reflectance Code (LaSRC) (Vermote et al., 2016). The Fmask algorithm (Zhu and Woodcock, 2012) was applied to detect cloud and cloud shadow in Landsat images. The L2SR products are spatially referenced using the Worldwide Reference System-2 (WRS-2) path rows and provided in the UTM projection. **Figure 1** illustrates the spatial and temporal distribution of all the L2SR images used in this study. As listed in **Table 1**, we used the blue, green, red, near infrared (NIR), and two shortwave infrared (SWIR1 and SWIR2) bands for the generation of SDC. Although Landsat sensors have a relatively narrow field of view (15 degrees), the Bidirectional Reflectance Distribution Function (BRDF) normalization was found to be effective to make multi-temporal Landsat observations more consistent (Claverie et al., 2015; Roy et al., 2016b). We applied the c-factor technique and global constant BRDF coefficients provided by Roy et al. (2016b) to obtain the Landsat Nadir BRDF adjusted Reflectance (NBAR) dataset for SDC generation.

Table 1: Attributes of Landsat 5 TM, 7 ETM+, 8-9 OLI, and MODIS Terra products (Markham and Helder, 2012; Masek et al., 2020; Morisette et al., 2002).

| | Wavelengths (micrometers) | | | | |
| --- | --- | --- | --- | --- | --- |
| | Landsat 5 TM | Landsat 7 ETM+ | Landsat 8 OLI | Landsat 9 OLI-2 | MODIS Terra |
| Blue | 0.450-0.520 | 0.450-0.515 | 0.452-0.512 | 0.452-0.512 | 0.459-0.479 |
| Green | 0.520-0.600 | 0.525-0.600 | 0.533-0.590 | 0.532-0.589 | 0.545-0.565 |
| Red | 0.630-0.690 | 0.630-0.690 | 0.636-0.673 | 0.636-0.672 | 0.620-0.670 |
| NIR | 0.760-0.900 | 0.760-0.900 | 0.851-0.879 | 0.850-0.879 | 0.841-0.876 |
| SWIR1 | 1.550-1.750 | 1.550-1.750 | 1.566-1.651 | 1.565-1.651 | 1.628-1.652 |
| SWIR2 | 2.080-2.350 | 2.080-2.350 | 2.107-2.294 | 2.105-2.294 | 2.105-2.155 |
| **Spatial resolution** | 30-m | 30-m | 30-m | 30-m | 500-m (near) |
| **Revisit frequency** | 16-day | 16-day | 16-day | 16-day | Daily (near) |



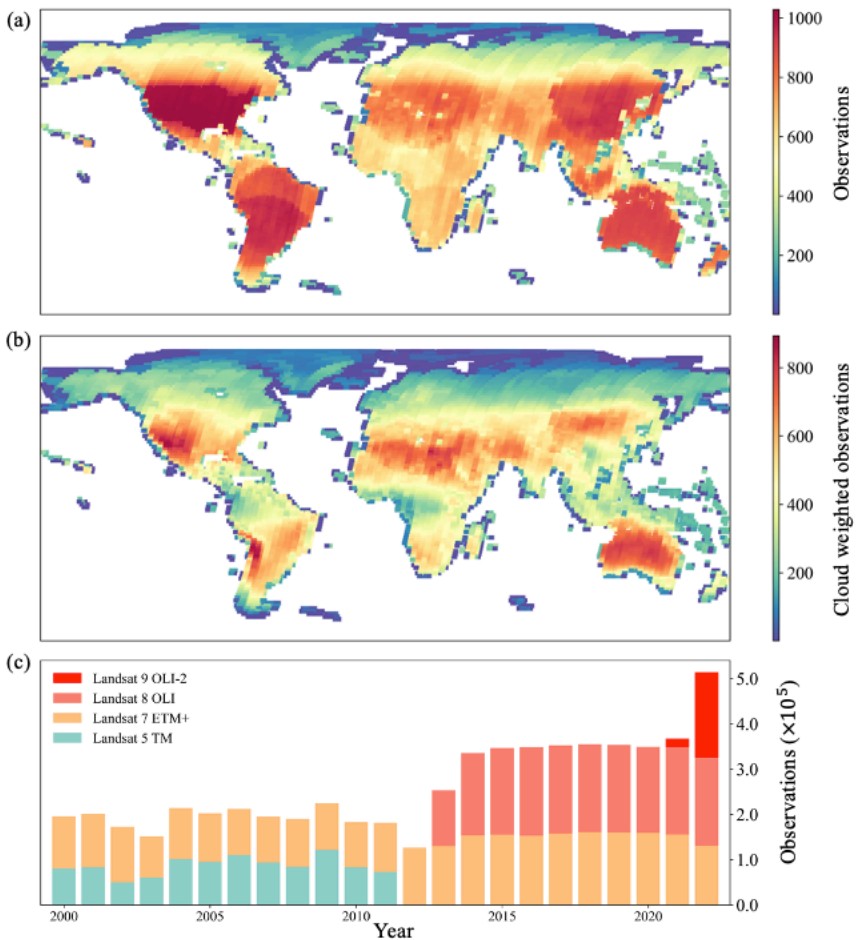

**Figure 1: Spatial and temporal distribution of L2SR images used in this study.**

## 2.2 MODIS Nadir BRDF-Adjusted Reflectance (NBAR) products

We reprocessed the MODIS MOD09GA Version 6.1 surface reflectance product to obtain a daily seamless (without missing

values) 500-m MODIS NBAR dataset, as detailed in Liang et al. (2024). The official MODIS MCD43A4 NBAR product was not employed due to persisting concerns associated with it, including the prevalence of missing data and residual influence of cloud and aerosols (Liang et al., 2024).

All MODIS Terra Surface Reflectance MOD09GA Version 6.1 images for the period 2000-2022 were acquired from NASA EarthData. The MOD09GA product is tiled in the MODIS sinusoidal system with a spatial resolution of about 500m.

A set of processing algorithms was applied to these MOD09GA images to derive a daily, 500-m resolution, seamless





MODIS NBAR data cube (Liang et al., 2024), including three main stages: (i) Land cover-based BRDF correction with the kernel-driven RossThick-LiSparse-Reciprocal (RTLSR) model using parameters derived from the MCD43A1 BRDF Model Parameters dataset and land cover maps from the MCD12Q1 land cover product; (ii) Outlier removal and gap filling using the ecosystem curve-fitting method; and (iii) Sliding-window temporal smoothing using the Savitzky-Golay filter. The generated MODIS NBAR seamless dataset provides seven spectral bands that are commonly used for terrestrial applications, six of which that have compatible bandwidths with Landsat sensors are employed for SDC generation, as listed in **Table 1**.

**2.3 The Harmonized Landsat and Sentinel-2 V2.0 surface reflectance product**

The NASA's Harmonized Landsat and Sentinel-2 (HLS) V2.0 products were used in the cross-comparison with the generated SDC product for quantitative assessment. The HLS products combine observations from the Landsat Operational Land Imager (OLI, since 2013) and the Sentinel-2 Multi-Spectral Instrument (MSI, since 2016), providing global surface reflectance data at a 30-m spatial resolution with a theoretical revisit interval of 2-3 days at the equator and even more frequent in areas of higher latitudes (Claverie et al., 2018). The creation of the HLS products involves four major processes (Claverie et al., 2018): (i) atmospheric correction and cloud masking, (ii) geometric resampling and geographic registration, (iii) BRDF normalization, and (iv) bandpass adjustment. The HLS products are gridded into the UTM Military Grid Reference System (MGRS) used by the Sentinel-2 products. The HLS S30 product (Sentinel-2-based 30-m product) is derived from 10/20-m Sentinel-2 bands using overlapping-area-weighted averaging, and the L30 product (Landsat-based 30-m product) is reprojected to the same Sentinel-2 grid using cubic convolution interpolation (Claverie et al., 2018). Both HLS L30 and S30 products are atmospherically corrected using the Land Surface Reflectance Code (LaSRC) (Vermote et al., 2016). The cloud mask used in the HLS products is a combination of the mask derived from the Fmask algorithm and the mask derived from the LaSRC algorithm. The Automated Registration and Orthorectification package (Gao et al., 2009) was employed to improve the co-registration accuracy of the HLS products. The HLS L30 and S30 products are delivered as NBAR, using the c-factor technique and global constant BRDF coefficients provided by Roy et al. (2016b). A bandpass adjustment is applied to S30 product using a global constant set of coefficients (Claverie et al., 2018).

**3 Methods**

**Figure 2** illustrates the overview of the SDC processing chain, comprising five key processing steps.



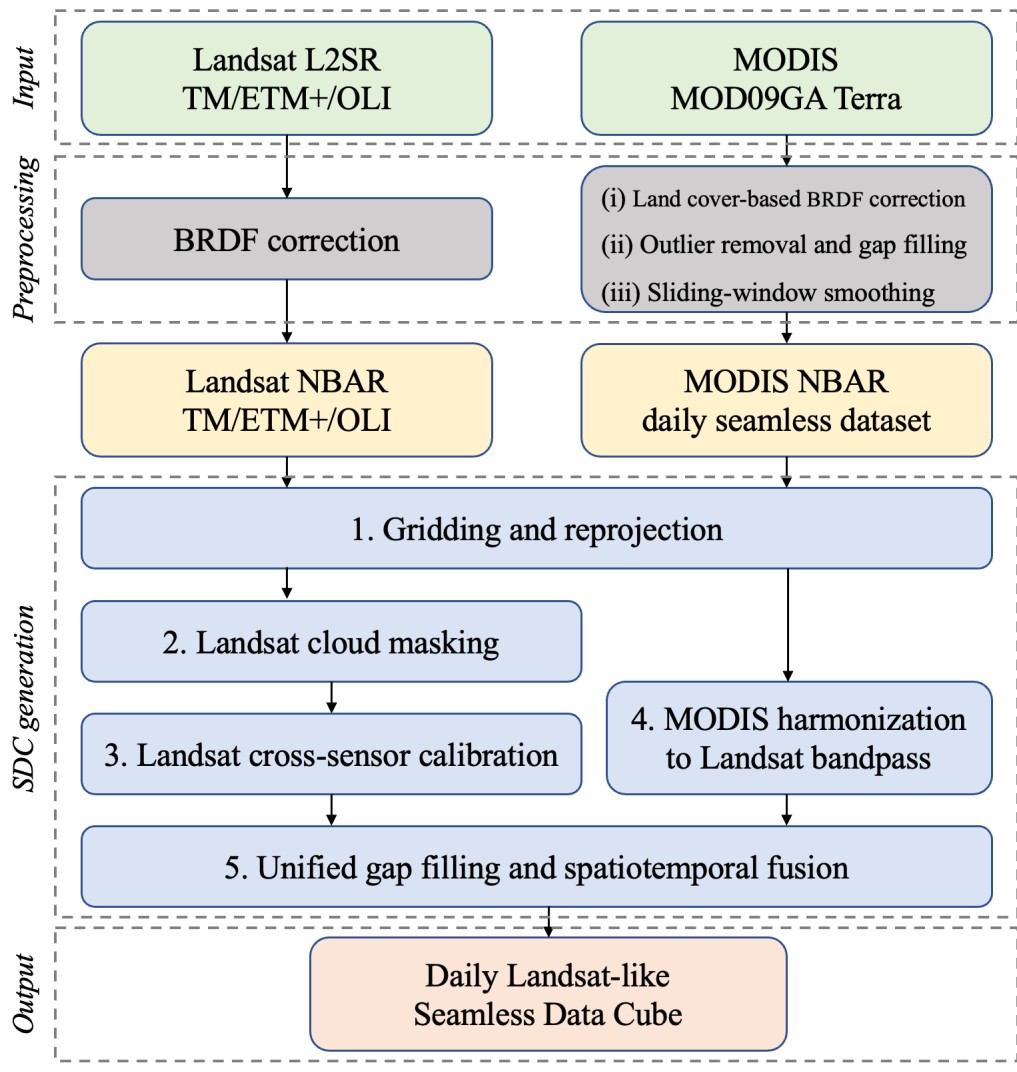

**Figure 2: Overview of the SDC processing chain for each task unit.**

## 3.1 Gridding and reprojection

The UTM-based Military Grid Reference System (MGRS) was chosen as the projection system for the SDC product, which was also adopted by ESA's Sentinel-2 products and NASA's HLS products (Claverie et al., 2018). It is noteworthy that our adopted grid slightly deviates from the Sentinel-2 grid. Since the original Landsat coordinate system exhibits a half-pixel (15 meters) offset relative to the Sentinel-2 grid, we expanded and shifted the original MGRS grid by 15 meters in each direction to align with the Landsat coordinate system. The SDC product is gridded into this modified MGRS system, with a tile size of 170 109.83×109.83 km (3661×3661 Landsat pixels). Although it has been found that the overlaps among neighbouring MGRS



tiles may result in resource wastage to some extent (Bauer-Marschallinger and Falkner, 2023), implementing this UTM-based projection system serves to minimize the need for resampling Landsat data, thereby reducing the introduction of additional errors (Dwyer et al., 2018).

The metadata for all Landsat and MODIS images has been pre-indexed into a database. For each SDC generation task unit, the metadata regarding all source data falling within specified spatial and temporal ranges can be efficiently retrieved from the database. Subsequently, all involved Landsat and MODIS source data are reprojected and gridded into the modified MGRS grid, using nearest neighbour resampling for Landsat and bilinear resampling for MODIS. The MODIS data are resampled to 30-m spatial resolution to streamline subsequent processing steps, and the computational costs for this upscaling operation are negligible. **Table 2** lists the input data products across distinct time periods utilized in SDC

generation.

**Table 2: Input product specifications across distinct periods.**

| Output | Period | Medium-resolution input | Coarse-resolution input |
|---|---|---|---|
| SDC | 2000-2011 | Landsat TM, ETM+ | MODIS NBAR |
| | 2012 | Landsat ETM+ | MODIS NBAR |
| | 2013-2022 | Landsat ETM+, OLI | MODIS NBAR |

## 3.2 Landsat cloud masking

Cloud and cloud shadow masks are essential for removing contaminated Landsat pixels in SDC generation. We used the Fmask (Zhu and Woodcock, 2012) detection results as primary cloud and cloud shadow indicators. There are still few clouds and heavy aerosols in Landsat images remain undetected by the current Fmask method, which may have noteworthy implications for subsequent data processing procedures (Chen et al., 2021). To mitigate this issue, an enhanced cloud filtering approach is employed to reduce residual clouds and cloud shadows in Landsat imagery, which comprises three

major steps:

(1) Firstly, the cloud and cloud shadow masks generated by the Fmask algorithm are expanded by a margin of 150 meters (Claverie et al., 2018). This dilation process is designed to exclude potentially contaminated pixels adjacent to the initially detected cloud and shadow areas.

(2) Secondly, a brightness-threshold filter combined with a spatial filter is applied to remove remaining highly reflective pixels, which is achieved by cross-comparing with MODIS NBAR data. This filter operates on a patch-wise basis, with each patch measuring 20×20 Landsat pixels. For a given image patch, we commence by computing the ratio of Landsat reflectance (summed over the six spectral bands) to MODIS reflectance for each pixel. Following this, if the median of all these Landsat-MODIS reflectance ratios within this image patch surpasses a predefined threshold, the entire image patch is





then flagged as cloudy. The threshold was set at 2 in this study, as this value effectively eliminates most residual clouds without being overly aggressive.

$$Median\left\{\frac{\sum_i \rho_i^{Landsat}(x,y)}{\sum_i \rho_i^{MODIS}(x,y)}, \dots, \text{for all pixels within the image patch}\right\} > 2, \tag{1}$$

where $(x, y)$ indicates the pixel location, $\rho_i^{Landsat}(x, y)$ and $\rho_i^{MODIS}(x, y)$ are the surface reflectance of band $i$ for the corresponding Landsat pixel and MODIS pixel, respectively.


(3) Further, a time-series outlier detection technique based on the Hampel filter combined with a spatial filter is employed to detect temporal outliers using the Vegetation Index (VI) of Landsat time series (Claverie et al., 2018).

$$VI(t) = \frac{\rho_{NIR}^{Landsat}(t)}{\rho_{Red}^{Landsat}(t)}, \tag{2}$$

where $t$ indicates the time phase, $\rho_{NIR}^{Landsat}(t)$ and $\rho_{Red}^{Landsat}(t)$ are the NIR and Red bands surface reflectance of Landsat at

time phase $t$.

For each sample $VI(t)$ of the Landsat time series, the Hampel filter computes the median of the VIs in a temporal window (center sample excluded).

$$VI_{median} = Median\{VI(t - \Delta t), \dots, VI(t - 1), VI(t + 1), \dots, VI(t + \Delta t)\}, \tag{3}$$

where $\Delta t$ represents the temporal window size.

Then, it estimates the Scale of Natural Variation (SNV) of each sample by deriving the median of the absolute deviations of the VIs in the temporal window from the median.

$$SNV = Median\{|VI(t - \Delta t) - VI_{median}|, \dots, |VI(t - 1) - VI_{median}|, |VI(t + 1) - VI_{median}|, \dots, |VI(t + \Delta t) - $$
$$VI_{median}|\} \tag{4}$$

If the center sample $VI(t)$ differs from the median $VI_{median}$ by more than five SNV, it is flagged as an outlier. No filter is applied if there are less than 3 samples within a 60-day temporal window. To eliminate isolated outlier pixels that generate a speckle effect, the sample pixel is flagged as an outlier only if the majority of its surrounding pixels are also flagged as outliers, cloud, or cloud shadow.




### 3.3 Landsat cross-sensor calibration

To ensure the temporal continuity of the generated SDC dataset, a Landsat cross-sensor calibration approach was employed to reduce the data inconsistencies between the input Landsat OLI and TM/ETM+ products. The linear regression models have been widely used to reduce cross-sensor reflectance difference (Chastain et al., 2019; Claverie, 2023; Claverie et al., 2018; Roy et al., 2016a; Shang and Zhu, 2019). It has been found that a single set of linear transformation coefficients are not proper for global-scale applications (Olthof and Fraser, 2024; Shang and Zhu, 2019). Therefore, our approach aims at building multiple transformation models for each MGRS tile and each spectral band separately.

The Landsat 7 ETM+ and Landsat 8 OLI share a sun-synchronous orbit at an altitude of approximately 710 km. Both satellites revisit a specific area on Earth every 16 days, with an 8-day offset from each other. Notably, there exists a discernible across-track overlap between each ETM+ observation and an adjacent OLI observation acquired one day apart (Roy et al., 2016a). Our cross-calibration approach uses matched (acquired one day apart) ETM+ and OLI observations in these overlapped regions to build linear regression models for each MGRS tile and each spectral band separately, adjusting ETM+ (and TM) observations to match the OLI spectral responses (Roy et al., 2016a).

For each MGRS tile, ETM+ and OLI observations in the overlap regions acquired only one day apart during 2014-2021 are reprojected into the modified MGRS grid using nearest neighbour resampling. Pixels flagged as cloud, cloud shadow, and snow/ice were discarded. Then, the remaining candidate pixels are used to build linear regression models to adjust reflectance differences for each spectral band. If the number of available pixels is insufficient to build the regression model, candidate pixels from adjacent MGRS tiles would be included. **Table 3** shows an example of calibration coefficients obtained at three MGRS tiles in this study.

$$\rho_i^{OLI} = a \times \rho_i^{ETM+} + b, \quad \text{for each band } i. \tag{5}$$

**Table 3: Calibration coefficients obtained at three MGRS tiles in this study.**

| Band | Tile 10SEH | Tile 49SGV | Tile 55HCC |
|---|---|---|---|
| Blue | OLI = 0.0021 + 0.9726 TM/ETM+ | OLI = 0.0089 + 0.8958 TM/ETM+ | OLI = 0.0056 + 0.9009 TM/ETM+ |
| Green | OLI = 0.0010 + 0.9856 TM/ETM + | OLI = 0.0064 + 0.9405 TM/ETM + | OLI = 0.0056 + 0.9344 TM/ETM + |
| Red | OLI = 0.0015 + 0.9719 TM/ETM + | OLI = 0.0050 + 0.9479 TM/ETM + | OLI = 0.0099 + 0.9232 TM/ETM + |
| NIR | OLI = 0.0062 + 0.9814 TM/ETM + | OLI = 0.0027 + 0.9839 TM/ETM + | OLI = 0.0317 + 0.8848 TM/ETM + |
| SWIR1 | OLI = 0.0039 + 0.9762 TM/ETM + | OLI = 0.0005 + 0.9887 TM/ETM + | OLI = 0.0354 + 0.8937 TM/ETM + |
| SWIR2 | OLI = 0.0019 + 0.9810 TM/ETM + | OLI = 0.0015 + 0.9811 TM/ETM + | OLI = 0.0306 + 0.8800 TM/ETM + |

### 3.4 MODIS harmonization to Landsat bandpass

Harmonizing MODIS to Landsat bandpass reduces inconsistencies between Landsat and MODIS observations, which has been proved effective to improve the reconstruction accuracy of subsequent gap filling and spatiotemporal fusion processes (Chen et al., 2023; Gevaert and García-Haro, 2015; Shi et al., 2022). Here, a cross-resolution data harmonization approach is



employed for harmonizing MODIS to Landsat OLI bandpass. This method involves utilizing matched Landsat and MODIS observations to establish multiple linear transformation models for each spectral band and each local image patch.

In contrast to previous methods that construct distinct transformation models for each land cover type (Cao et al., 2020; Shen et al., 2013; Yang et al., 2020), our approach adopts a patch-wise harmonization strategy with an overlapping
mechanism to tackle spatial heterogeneity. This strategy avoids the necessity for high-accuracy land cover maps while concurrently ensuring computational efficiency. As illustrated in **Figure 3**, Landsat images are paired with MODIS images acquired on corresponding dates, with the exclusion of contaminated Landsat pixels such as clouds and cloud shadows. Subsequently, for each image patch and each spectral band, a linear transformation model is constructed utilizing candidate pixels within that image patch. The MODIS reflectance is then adjusted using the obtained transformation coefficients to
generate more "harmonized" MODIS data. In areas of patch overlap, the transformation coefficients are averaged for the final adjustments.

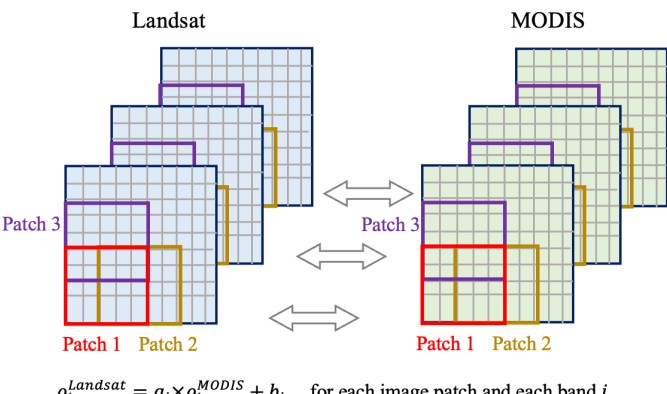

$$\rho_i^{Landsat} = a_i \times \rho_i^{MODIS} + b_i, \quad \text{for each image patch and each band } i$$

**Figure 3: An illustration of patch-wise harmonization with overlapping.**


### 3.5 Unified gap filling and spatiotemporal fusion

Existing spatiotemporal fusion algorithms generally require cloud-free seamless Landsat images as input (Gao et al., 2006; Shi et al., 2022; Zhu et al., 2016, 2010), which may harm their data efficiency and performances, especially in cloudy areas where there are few cloud-free Landsat images available. Therefore, previous studies (Chen et al., 2018; Liu et al., 2021)
applied gap filling algorithms to partly contaminated Landsat images first, and then use these gap-filled images for subsequent fusion process. Different from these approaches, we propose the Unified, ROBust, OpTimization-based spatiotemporal reconstruction model (uROBOT), which can tackle both gap filling and spatiotemporal fusion problems in a unified manner.





As shown in **Figure 4**, the input data for uROBOT consist of a matched time series of Landsat-MODIS image patches $D_F$ and $D_C$ acquired at $\{T_1, \dots, T_n\}$, a MODIS image patch $C_p$ acquired at the prediction phase $T_p$, and a partially contaminated Landsat image $F_p$ acquired at $T_p$ (case 1 in **Figure 4**). The output is the predicted reflectance values for the unobserved segments of the Landsat image $F_p^-$ at $T_p$. Should $F_p$ be entirely contaminated/unobserved, the scenario simplifies to a conventional spatiotemporal fusion problem (case 2 in **Figure 4**).

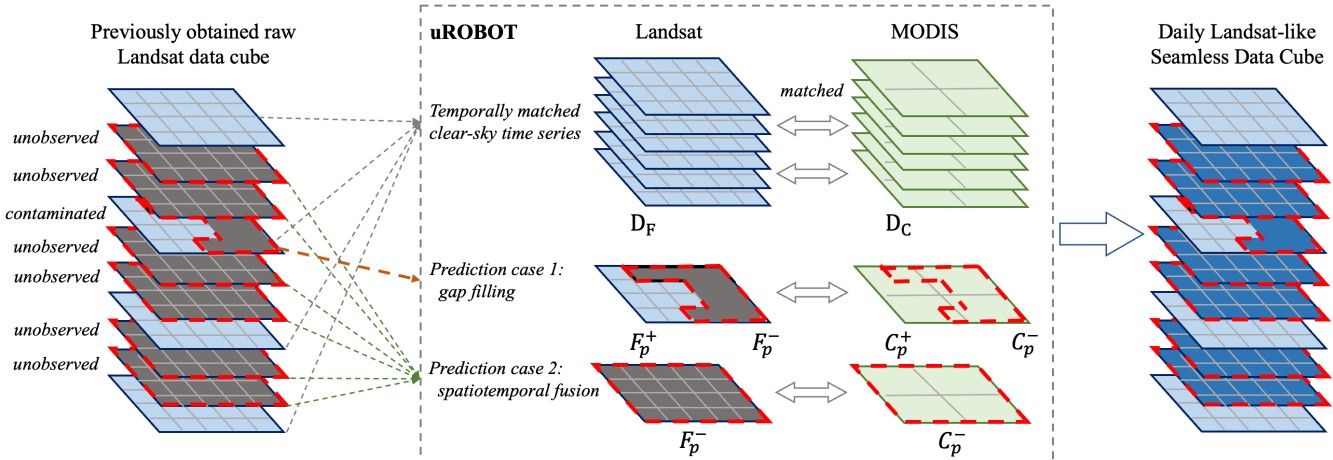


**Figure 4: Input/output settings of uROBOT. The superscripts "+" and "-" indicate the corresponding observed and unobserved (or contaminated) segments of Landsat image at the prediction phase. The uROBOT can tackle both gap filling (case 1) and spatiotemporal fusion (case 2) problems in a unified manner.**

**3.5.1 Preliminaries of the uROBOT model**

Similar to the previous spatiotemporal fusion model (Chen et al., 2023), the basic assumption of uROBOT is that the MODIS image $C_p$ can be accurately approximated by a linear combination of other similar MODIS images in the input time-series data, given by

$$C_p \approx D_c \alpha, \tag{6}$$

where $C_p$ is the MODIS image patch acquired at the prediction phase $T_p$,

$D_C$ is a matrix that stacks the input MODIS time-series image patches acquired at $\{T_1, \dots, T_n\}$,

$\alpha$ is a sparse vector that selects out MODIS image patches similar to $C_p$ and combines them to approximate $C_p$.

Then, it is assumed that the representation coefficients $\alpha$ can be transferred to corresponding Landsat images, and obtain

an estimation for $F_p$





$$\hat{F}_p = D_F \alpha, \tag{7}$$

where $\hat{F}_p$ is the estimated Landsat image patch at $T_p$,

$D_F$ is a matrix that stacks the input Landsat time-series image patches acquired at $\{T_1, \dots, T_n\}$.

The representation coefficients $\alpha$ can be obtained by solving an optimization problem with two extra regularization terms. Detailed explanations regarding these two regularization terms will be provided subsequently.

$$\min_{\alpha} |C_p - D_C \alpha|_2^2 + \lambda|\alpha|_1 + \beta(|F_{interp} - D_F \alpha|_2^2) + \mu(|F_p^+ - D_F^+ \alpha|_2^2), \tag{8}$$

where $\lambda, \beta, \mu$ are scalars,

$\alpha$ is a vector that flattens the representation coefficients,

$C_p$ is a vector that flattens the MODIS image patch at $T_p$,

$D_C$ is a matrix that stacks the MODIS time-series image patches acquired at $\{T_1, \dots, T_n\}$,

$D_F$ is a matrix that stacks the Landsat time-series image patches acquired at $\{T_1, \dots, T_n\}$,

$F_{iterp}$ is a vector that flattens an interpolated Landsat image patch at $T_p$ (further elucidation on this matter will be presented subsequently),

$F_p^+$ is a vector that flattens the observed part of $F_p$,

and $D_F^+$ is a matrix that stacks the corresponding observed parts of $D_F$.

Beyond the previous spatiotemporal fusion model (Chen et al., 2023), the uROBOT model accepts the observed part $F_p^+$ (if there is, as case 1 in **Figure 4**) as an extra input. Hence, the regularization term $\mu(|F_p^+ - D_F^+ \alpha|)$ allows uROBOT to

exploits the observed segments $F_p^+$ to better reconstruct the target image $F_p$. This feature enables the uROBOT model to handle both gap filling and spatiotemporal fusion problems in a unified manner.

The $F_{interp}$ is an interpolated Landsat image patch at $T_p$, obtained using the combination of nearing-date Landsat observations weighted by corresponding MODIS data. Therefore, the regularization term $\beta(|F_{interp} - D_F \alpha|_2^2)$ serves to enhance the temporal continuity of the final predicted Landsat image $\hat{F}_p = D_F \alpha$.

To handle extreme conditions such as ephemeral land cover changes, the uROBOT model also distributes the approximation residuals into the prediction (Chen et al., 2023), and the final prediction is formulated as

$$\hat{F}_p = D_F \alpha + (C_p - D_C \alpha), \tag{9}$$



### 3.5.2 Implementation of uROBOT for SDC reconstruction

As shown in **Figure 4**, the uROBOT model reconstructs seamless Landsat images for each prediction phase separately. At
each prediction phase $T_p$, the uROBOT model takes three main steps to reconstruct the corresponding Landsat image $F_p$:

(1) Firstly, the uROBOT model constructs an interpolated image $F_{interp}$ using the weighted combination (Zhu et al.,
2010) of clear-sky Landsat pixels acquired nearest to the prediction phase $T_p$.

$$F_{interp}(x,y) = w_1 \times F_1(x,y) + w_2 \times F_2(x,y), \qquad \text{for each pixel location } (x,y) \tag{10}$$

where $(x,y)$ indicates a given pixel location,

$F_1(x,y)$ is the cloud-free Landsat pixel acquired nearest to and before $T_p$,

$F_2(x,y)$ is the cloud-free Landsat pixel acquired nearest to and after $T_p$,

and the weights $w_1$ and $w_2$ are obtained using corresponding MODIS pixels.

$$w_1 = \frac{(C_2(x,y)-C_p(x,y))^2}{(C_1(x,y)-C_p(x,y))^2+(C_2(x,y)-C_p(x,y))^2} \quad \text{and.} \quad w_2 = \frac{(C_1(x,y)-C_p(x,y))^2}{(C_1(x,y)-C_p(x,y))^2+(C_2(x,y)-C_p(x,y))^2}, \tag{11}$$

where $C_p(x,y)$ is the corresponding MODIS pixel acquired at $T_p$,

$C_1(x,y)$ is the corresponding cloud-free MODIS pixel acquired on the same date as $F_1(x,y)$,

and $C_2(x,y)$ is the corresponding cloud-free MODIS pixel acquired on the same date as $F_2(x,y)$.

(2) Secondly, the uROBOT model utilizes all input time series data and the spatial information of $F_p^+$ to do similar
image matching and approximation (Chen et al., 2023), by solving the optimization problem in **Equation (8)** to obtain the
representation coefficients $\alpha$.

(3) Then, the last step is to reconstruct the target $F_p^-$ using the obtained coefficients $\alpha$ and all input data, as

$$\hat{F}_p^- = D_F^- \alpha + (C_p^- - D_C^- \alpha), \tag{12}$$

where $\hat{F}_p^-$ is an estimation of the unobserved/contaminated segments of $F_p$,

$D_F^-$, $C_p^-$, and $D_C^-$ are the corresponding masked parts of $D_F$, $C_p$, and $D_C$.

### 3.6 Quantitative assessment using the leave-one-out approach

The leave-one-out assessment approach has been widely used to evaluate the accuracy of reconstructed surface reflectance values in previous gap filling and spatiotemporal fusion studies (Chen et al., 2011; Gao et al., 2006; Zhu et al., 2016, 2010). This approach initially excludes a certain Landsat image from the input data, subsequently using the remaining input data to reconstruct the excluded image and thereafter evaluating discrepancies between the originals and reconstructions using standard metrics, such as Correlation Coefficient (CC), Root Mean Square Error (RMSE), Mean Absolute Error (MAE), and rMAE (MAE normalized by true surface reflectance values).

A total of 425 test sites that distributed across the globe are selected randomly, each of which covers a 6 km × 6 km area. These test sites were grouped by their dominant land cover types using the FROM_GLC land cover map (Gong et al., 2013; C. Li et al., 2017). **Figure 5** shows the spatial distribution and corresponding dominant land cover types of the 425 global test sites. For each Landsat image at each test site, we exclude it from the input data and applied SDC reconstruction with the remaining data, separately. Then, the reconstruction accuracy is evaluated by comparing the reconstructed images with the original Landsat images. The accuracy assessment is conducted in four years (2001, 2004, 2012, 2021) representative of different input data conditions.

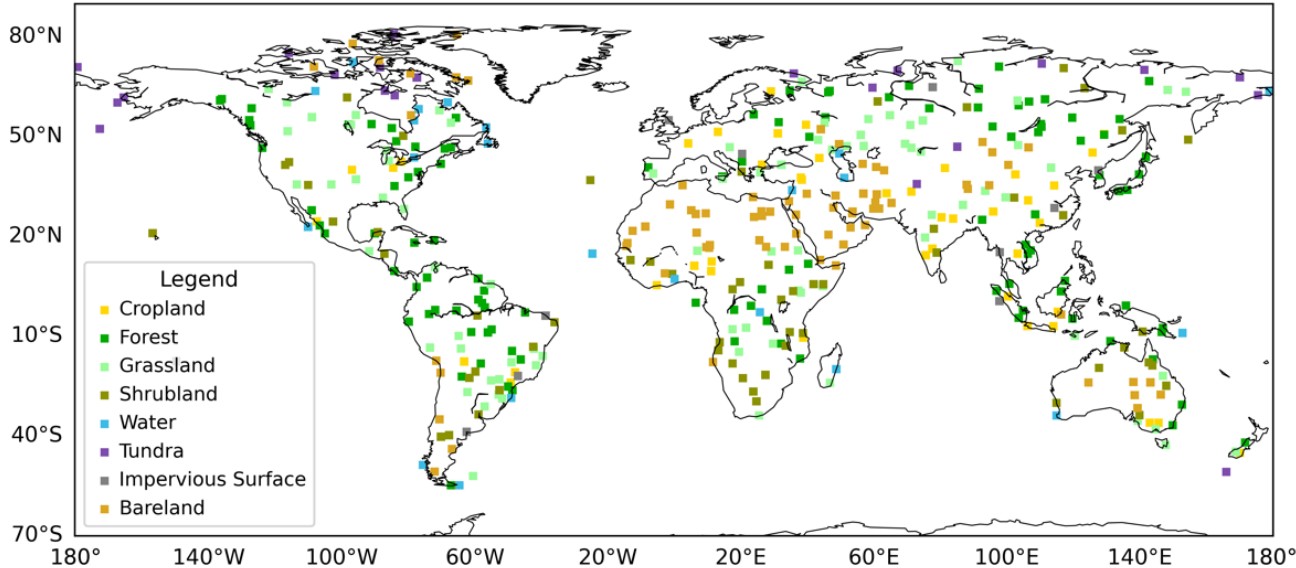

**Figure 5: Spatial distribution and corresponding dominant land cover types of the 425 global test sites.**



**3.7 Quantitative assessment by cross-comparing with the HLS products**

Another assessment approach is to cross-compare the gap-filled SDC dataset with actual observations from other sensors. The NASA's HLS products provide dense 30-m observation data by harmonizing Landsat OLI (since 2014) and Sentinel-2 MSI (since 2016) products, making itself a good reference data to evaluate the SDC product for the period 2016-2022. We 375   selected 22 MGRS tiles representative of different land cover types as test sites (Chen et al., 2023) and evaluated the agreement between the SDC and HLS products in the year 2021. **Figure 6** shows the spatial distribution of the 22 MGRS tiles in the cross-comparison. Each MGRS tile covers a 109.8×109.8 km area. Since the L30 product are derived from Landsat OLI data, we employed the leave-one-out validation strategy (the same as in last section) for the cross-comparison with L30 data. The least square regression method was used to reduce spectral bandpass differences between the SDC and 380   HLS data for each spectral band and each test site.

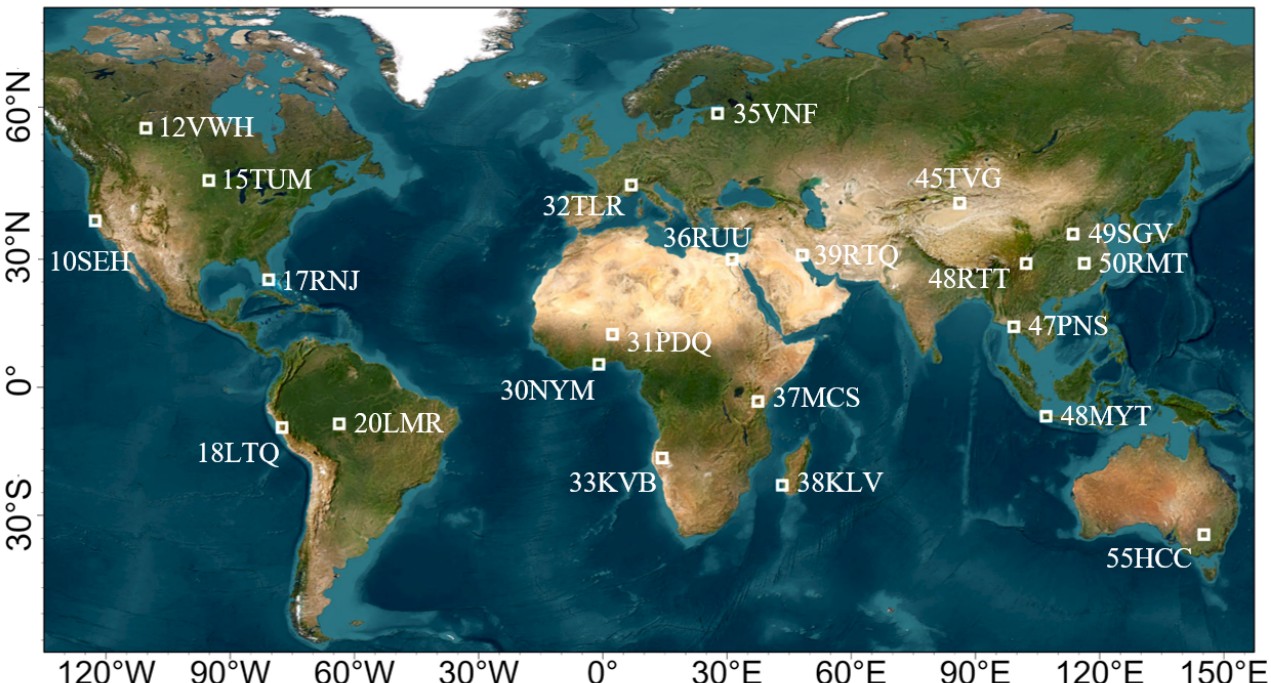

**Figure 6: Spatial distribution of the 22 MGRS tiles involved in the cross-comparison with HLS.**




# 4 Results

## 4.1 Global 30-m daily Seamless Data Cube (SDC) of land surface reflectance

Based on the developed framework, a global, 30-m, 23-year (2000-2022), and daily surface reflectance SDC dataset was generated by combining multi-sensor observations from Landsat TM/ETM+/OLI and MODIS Terra products. The generated
SDC dataset is tiled into the modified MGRS grid, as described in **Section 3.1**. This gridding system includes 18,466 modified MGRS tiles (each of which includes 3661×3661 Landsat pixels), covering most of global land surface except Antarctica, as shown in **Figure 7**.

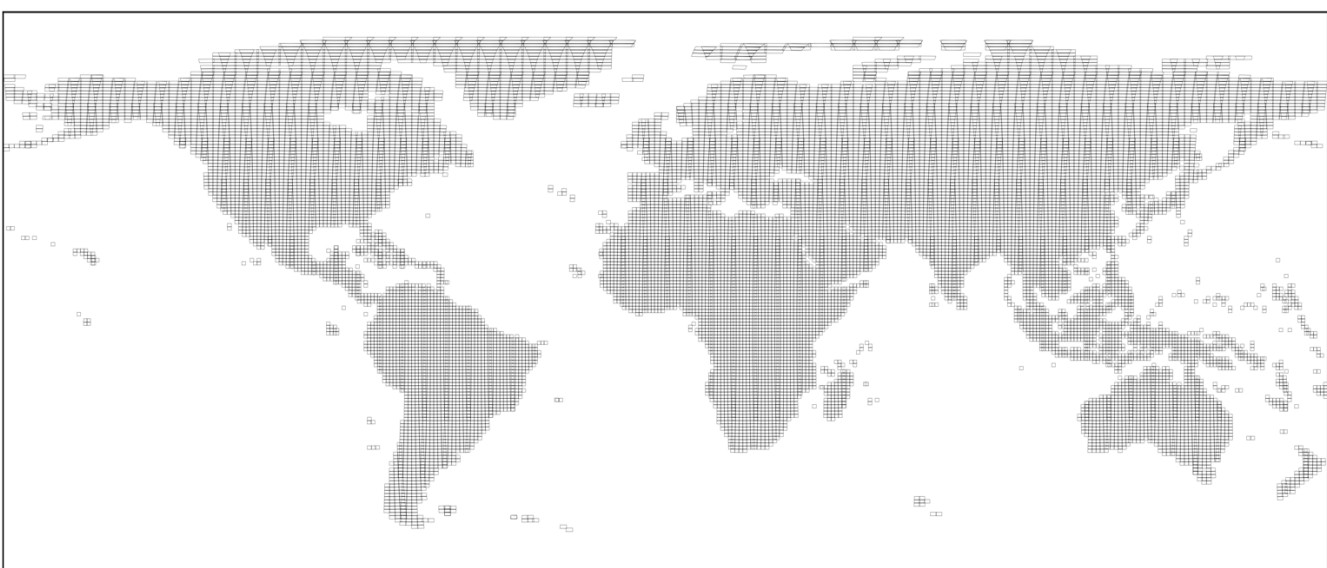

**Figure 7: Distribution of the 18,466 modified MGRS tiles used for SDC generation.**


**Figs. 8-11** depict four examples of SDC time series in comparison with the HLS data (not used in the generation of SDC), with a specific focus on land cover changes. **Figure 8** illustrates a crop harvest event that took place in April 2021 in Egypt. Remarkably, the SDC time series demonstrates a more adept representation of the various crop harvest stages compared to the L30 time series. This enhanced performance is attributed to the incorporation of more frequent temporal information
from MODIS. **Figure 9** presents the second case in the Poyang Lake wetland, a region prone to frequent cloud cover. Although there are only limited cloud-free Landsat observations, the temporal phases of land-water transition are effectively captured in the SDC time series. The third case typified a tundra region in Canada. As shown in **Figure 10**, the SDC time series accurately reflects the snow season and vegetation growth in the tundra ecosystem. The fourth case in **Figure 11** illustrates a flood event that occurred in India. This region experienced flooding in October 2013, a period when Sentinel-2 data was unavailable. During December, the water gradually receded, as depicted in the figure. The results demonstrate the

proficiency of the SDC data in monitoring rapid land cover changes. Further, the SDC dataset exhibits robust consistency in both spatial and temporal dimensions, with an extended temporal coverage dating back to 2000.

**Figure 8: False-color composites of SDC and HLS data, and time series of red SR (red) and NIR SR (blue) of the central pixel located at 30.4851º N, 31.9383º E (Egypt, tile 36RUU) from Feb-15 to Jul-15 2021.**



**Figure 9: False-color composites of SDC and HLS data, and time series of red SR (red) and NIR SR (blue) of the central pixel located at 29.0965º N, 116.1107º W (China, tile 50RMT) from Feb-15 to Jul-15 2021.**




**Figure 10: False-color composites of SDC and HLS data, and time series of red SR (red) and NIR SR (blue) of the central pixel located at 56.2778º N, 110.9034º W (Canada, tile 12VWH) from Feb-15 to Jun-15 2021.**




**Figure 11: False-color composites of SDC and HLS data, and time series of red SR (red) and NIR SR (blue) of the central pixel located at 23.2923º N, 68.7947º E (India, tile 42QVL) from Nov-15 2013 to Mar-15 2014.**




## 4.2 Accuracy assessment results using the leave-one-out approach

**Table 4** displays the results of quantitative assessments for the 425 global test sites across four typical years, utilizing different input data settings. Notably, the SDC attains its peak reconstruction accuracy in 2021, when incorporating Landsat ETM+, OLI, and MODIS data. The reconstruction accuracy of SDC in 2012 is comparatively diminished in 2012, when

Landsat TM and OLI observations are not available. **Table 5** illustrates the SDC reconstruction accuracy across test sites characterized by different land cover types. The reconstruction accuracy of SDC in water bodies and tundra areas is observed to manifest relatively higher error levels. **Figure 12** displays scatter plots depicting the predicted SDC and actual Landsat surface reflectance values in the leave-one-out assessment. The majority of data points closely align with the 1:1 line, indicating a robust consistency between the predicted and actual reflectance values. Overall, these results substantiate that

reconstructed SDC surface reflectance values achieve a high level of accuracy on a global scale.

**Table 4: Accuracy of SDC reconstruction at the 425 global test sites (ETM+*: scan-line corrector failure).**

| Band | 2001 (TM and ETM+ with MODIS) | | | 2004 (TM and ETM+* with MODIS) | | | 2012 (ETM+* with MODIS) | | | 2021 (ETM+* and OLI with MODIS) | | |
|------|------|------|------|------|------|------|------|------|------|------|------|------|
| | RMSE | MAE | CC | RMSE | MAE | CC | RMSE | MAE | CC | RMSE | MAE | CC |
| Blue | 0.016 | 0.012 | 0.731 | 0.015 | 0.011 | 0.757 | 0.016 | 0.011 | 0.781 | 0.015 | 0.011 | 0.779 |
| Green | 0.017 | 0.012 | 0.791 | 0.015 | 0.011 | 0.817 | 0.015 | 0.011 | 0.829 | 0.016 | 0.012 | 0.847 |
| Red | 0.019 | 0.014 | 0.825 | 0.017 | 0.013 | 0.851 | 0.018 | 0.013 | 0.858 | 0.017 | 0.013 | 0.875 |
| NIR | 0.028 | 0.021 | 0.873 | 0.026 | 0.019 | 0.893 | 0.026 | 0.019 | 0.894 | 0.027 | 0.020 | 0.904 |
| SWIR1 | 0.024 | 0.018 | 0.818 | 0.022 | 0.016 | 0.835 | 0.021 | 0.016 | 0.851 | 0.021 | 0.016 | 0.870 |
| SWIR2 | 0.018 | 0.013 | 0.795 | 0.016 | 0.012 | 0.826 | 0.017 | 0.012 | 0.837 | 0.016 | 0.012 | 0.845 |


**Table 5: Accuracy of SDC reconstruction at the 425 global test sites for different land cover types.**

| Band | Cropland | | | | Forest | | | | Grassland | | | | Shrubland | | | |
|------|------|------|------|---------|------|------|------|---------|------|------|------|---------|------|------|------|---------|
| | RMSE | MAE | CC | rMAE (%) | RMSE | MAE | CC | rMAE (%) | RMSE | MAE | CC | rMAE (%) | RMSE | MAE | CC | rMAE (%) |
| Blue | 0.012 | 0.009 | 0.763 | 13% | 0.014 | 0.010 | 0.684 | 17% | 0.015 | 0.011 | 0.808 | 11% | 0.011 | 0.008 | 0.758 | 11% |
| Green | 0.012 | 0.010 | 0.812 | 9% | 0.013 | 0.010 | 0.750 | 11% | 0.015 | 0.011 | 0.858 | 8% | 0.011 | 0.008 | 0.826 | 8% |
| Red | 0.015 | 0.012 | 0.866 | 11% | 0.015 | 0.011 | 0.777 | 13% | 0.017 | 0.013 | 0.896 | 9% | 0.013 | 0.009 | 0.875 | 9% |
| NIR | 0.027 | 0.020 | 0.892 | 7% | 0.029 | 0.022 | 0.880 | 8% | 0.026 | 0.019 | 0.907 | 7% | 0.021 | 0.016 | 0.898 | 6% |
| SWIR1 | 0.026 | 0.019 | 0.867 | 8% | 0.020 | 0.015 | 0.820 | 10% | 0.022 | 0.017 | 0.863 | 8% | 0.020 | 0.015 | 0.863 | 7% |
| SWIR2 | 0.021 | 0.016 | 0.882 | 10% | 0.013 | 0.010 | 0.766 | 11% | 0.018 | 0.013 | 0.856 | 9% | 0.016 | 0.012 | 0.870 | 8% |





| Band | Water | | | | Tundra | | | | Impervious surface | | | | Bareland | | | |
|------|------|-----|-----|-----------|------|-----|-----|-----------|------|-----|-----|-----------|------|-----|-----|-----------|
| | RMSE | MAE | CC | rMAE (%) | RMSE | MAE | CC | rMAE (%) | RMSE | MAE | CC | rMAE (%) | RMSE | MAE | CC | rMAE (%) |
| Blue | 0.023 | 0.017 | 0.778 | 15% | 0.049 | 0.033 | 0.921 | 15% | 0.012 | 0.009 | 0.726 | 13% | 0.012 | 0.009 | 0.784 | 6% |
| Green | 0.023 | 0.017 | 0.829 | 12% | 0.048 | 0.033 | 0.927 | 13% | 0.013 | 0.009 | 0.792 | 10% | 0.013 | 0.010 | 0.858 | 5% |
| Red | 0.025 | 0.018 | 0.834 | 14% | 0.051 | 0.035 | 0.930 | 13% | 0.015 | 0.011 | 0.836 | 12% | 0.016 | 0.012 | 0.877 | 4% |
| NIR | 0.033 | 0.024 | 0.858 | 13% | 0.058 | 0.043 | 0.931 | 12% | 0.027 | 0.019 | 0.906 | 8% | 0.018 | 0.014 | 0.880 | 4% |
| SWIR1 | 0.020 | 0.015 | 0.818 | 15% | 0.035 | 0.026 | 0.795 | 22% | 0.022 | 0.016 | 0.883 | 9% | 0.021 | 0.015 | 0.847 | 6% |
| SWIR2 | 0.016 | 0.012 | 0.778 | 15% | 0.026 | 0.020 | 0.714 | 20% | 0.017 | 0.013 | 0.880 | 11% | 0.016 | 0.012 | 0.856 | 6% |

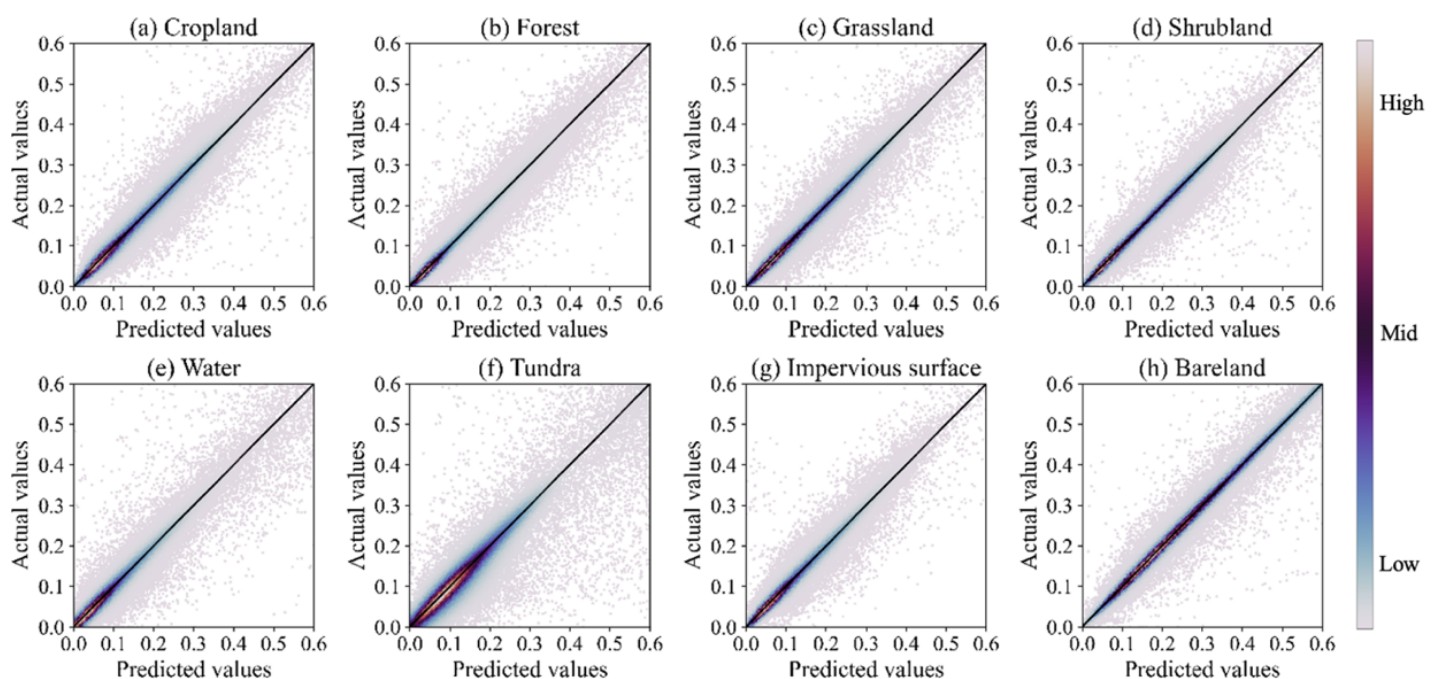

**Figure 12: Scatter plots of the actual and predicted values for the test sites of different land cover types.**


### 4.3 Accuracy assessment results by cross-comparing with the HLS products

**Table 6** presents the quantitative cross-comparison results between the SDC and HLS L30/S30 products at the 22 MGRS tiles. The results indicate a higher level of agreement between the SDC and HLS L30 products compared to that between the



SDC and HLS S30 products. This discrepancy may be attributed to the spectral differences between the Landsat OLI and
Sentinel-2 MSI instruments. Notably, since the metric values listed in **Table 6** are calculated for test sites at a different
spatial scale (109.8 km×109.8 km here, and 6 km×6 km in Section 4.2), the listed RMSD and CC values are not directly
comparable to those in **Tables 4-5**. As described in **Section 3.1**, the gridding system of SDC has a 15-meter offset compared
to the Sentinel-2 gridding system used by the HLS products. This 15-meter offset may also cause systematic deviations in
the cross-comparison.

**Figure 13** presents scatter plots depicting the reconstructed SDC data in comparison to HLS data. The blue band in the
plots indicates higher deviations, known to be sensitive to atmospheric conditions. Most data points lie near the 1:1 line,
indicating a high degree of agreement between the SDC and HLS products.

**Table 6: Cross-comparison results of between the SDC with HLS L30 and S30 products at the 22 MGRS tiles.**

| Band | Compare SDC with HLS L30 | | | Compare SDC with HLS S30 | | |
|---|---|---|---|---|---|---|
| | RMSD | MAD | CC | RMSD | MAD | CC |
| Blue | 0.058 | 0.014 | 0.892 | 0.059 | 0.017 | 0.849 |
| Green | 0.055 | 0.015 | 0.906 | 0.060 | 0.020 | 0.863 |
| Red | 0.056 | 0.017 | 0.920 | 0.060 | 0.022 | 0.890 |
| NIR | 0.054 | 0.023 | 0.924 | 0.059 | 0.027 | 0.899 |
| SWIR1 | 0.030 | 0.017 | 0.977 | 0.034 | 0.022 | 0.971 |
| SWIR2 | 0.025 | 0.014 | 0.980 | 0.028 | 0.018 | 0.974 |






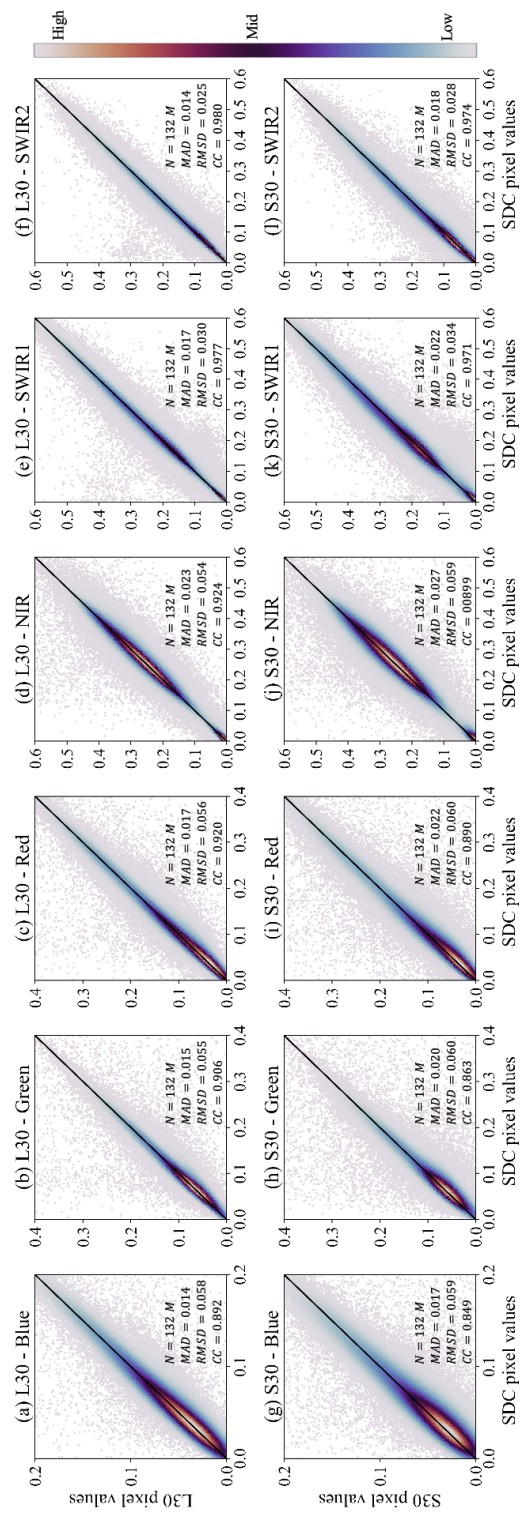

Figure 13: Scatter plots of SDC and HLS reflectance values in the cross-comparison.



## 4.4 The effectiveness of Landsat cross-sensor calibration

The evaluation of our Landsat cross-sensor calibration method involved a comparison of surface reflectance differences
before and after the bandpass adjustment. The surface reflectance differences are quantified by the RMSDs between ETM+
and OLI observations (temporally interpolated to align with the acquisition dates of ETM+). This comparison is conducted
for six spectral bands across six MGRS tiles (10SEH, 17RKQ, 32UNA, 36MWS, 49SGV, and 55HCC). **Figure 14** presents
the surface reflectance differences before and after the cross-calibration process for six spectral bands at the six test sites.
Additionally, **Figure 15** illustrates two examples of calibrated blue and NIR surface reflectance time series. The results
demonstrate the effectiveness of our method in reducing the surface reflectance differences between ETM+ and OLI
observations, aligning them more closely using obtained local-scale linear transformation models.

|  | Blue | Green | Red | NIR | SWIR1 | SWIR2 |
|---|---|---|---|---|---|---|
| Before | 86.9 | 76.8 | 111.3 | 215.4 | 186.6 | 145.4 |
| After | 47.9 | 54.9 | 82.2 | 160.1 | 151.4 | 117.2 |
| Reduce | 44.8% | 28.5% | 26.1% | 25.7% | 18.9% | 19.4% |

**10SEH**

|  | Blue | Green | Red | NIR | SWIR1 | SWIR2 |
|---|---|---|---|---|---|---|
| Before | 113.7 | 121.8 | 174.8 | 388.1 | 343.6 | 333.8 |
| After | 59.4 | 80.0 | 107.0 | 242.0 | 199.3 | 193.5 |
| Reduce | 47.8% | 34.3% | 38.8% | 37.6% | 42.0% | 42.0% |

**17RKQ**

|  | Blue | Green | Red | NIR | SWIR1 | SWIR2 |
|---|---|---|---|---|---|---|
| Before | 142.8 | 158.0 | 180.4 | 495.0 | 269.6 | 248.5 |
| After | 62.2 | 66.2 | 112.0 | 351.2 | 172.4 | 156.7 |
| Reduce | 56.4% | 58.1% | 37.9% | 29.1% | 36.1% | 36.9% |

**32UNA**

|  | Blue | Green | Red | NIR | SWIR1 | SWIR2 |
|---|---|---|---|---|---|---|
| Before | 83.6 | 73.9 | 98.3 | 230.6 | 174.1 | 142.5 |
| After | 43.8 | 54.8 | 82.3 | 158.1 | 150.4 | 128.4 |
| Reduce | 47.6% | 25.8% | 16.3% | 31.4% | 13.6% | 9.9% |

**36MWS**

|  | Blue | Green | Red | NIR | SWIR1 | SWIR2 |
|---|---|---|---|---|---|---|
| Before | 189.1 | 165.2 | 182.8 | 316.7 | 187.3 | 195.4 |
| After | 75.4 | 72.0 | 103.3 | 259.6 | 123.3 | 138.9 |
| Reduce | 60.1% | 56.4% | 43.5% | 18.0% | 34.2% | 28.9% |

**49SGV**

|  | Blue | Green | Red | NIR | SWIR1 | SWIR2 |
|---|---|---|---|---|---|---|
| Before | 96.9 | 113.1 | 216.2 | 267.3 | 406.7 | 421.7 |
| After | 68.8 | 88.9 | 163.6 | 176.9 | 303.0 | 312.0 |
| Reduce | 29.0% | 21.4% | 24.3% | 33.8% | 25.5% | 26.0% |

**55HCC**

**Figure 14: The mean RMSDs between ETM+ and OLI observations before and after the cross-calibration for six spectral bands at six MGRS tiles.**

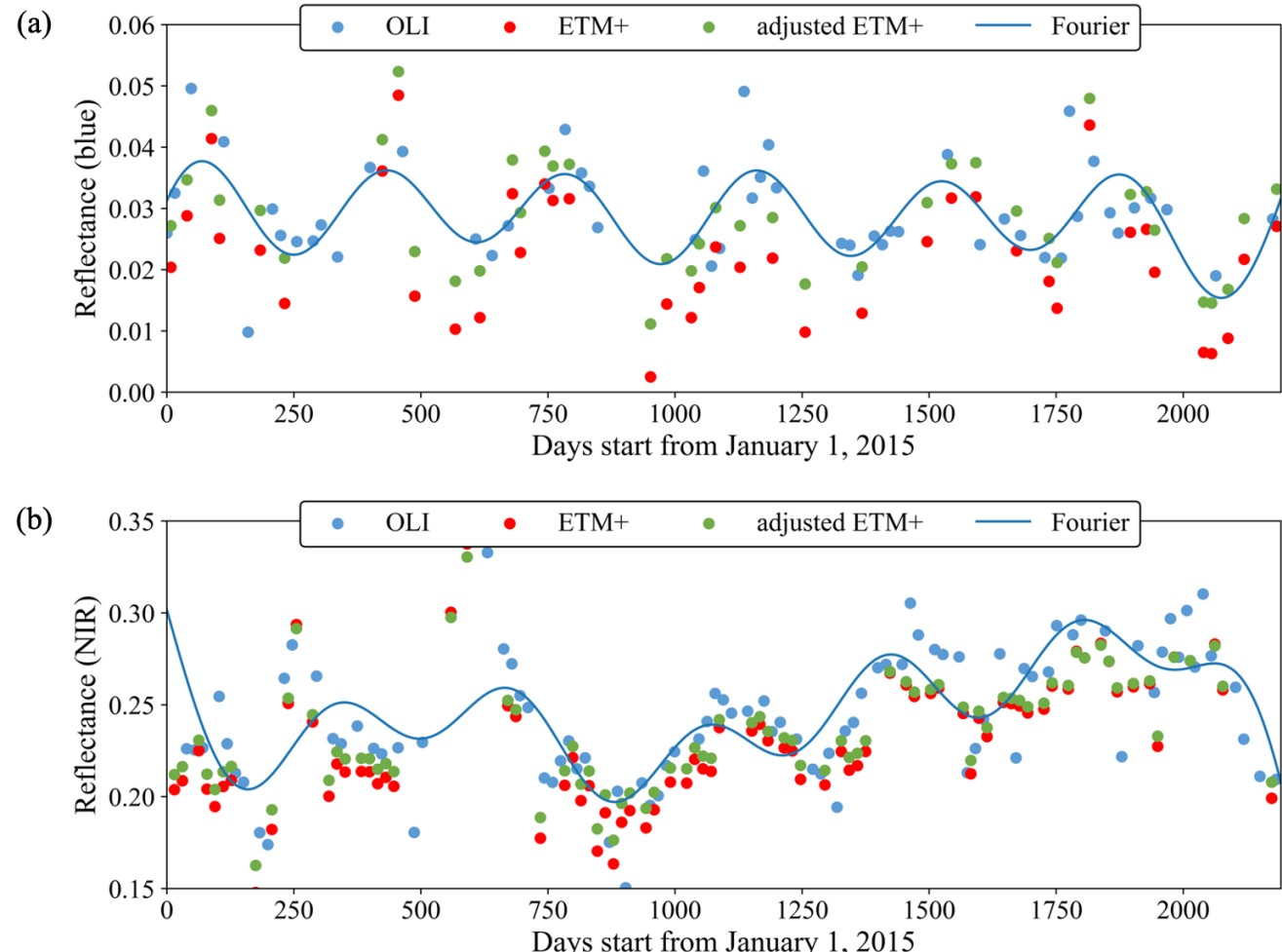

Figure 15: Time series of original and calibrated surface reflectance: (a) located at 23.2923º N, 68.7947º E; (b) located at 23.2923º N, 68.7947º E. The blue lines are the estimated curves of the OLI observations based on the Fourier approach (Dash et al., 2010; Shang and Zhu, 2019).

## 4.5 The effects of remaining Landsat sensor differences

The effects of remaining Landsat sensor differences on the SDC reconstruction accuracy were investigated in this section. We evaluated the reconstruction accuracy with different input data settings in three typical years (2001, 2004, 2021) using the 425 global test sites from Section 3.6. The accuracy assessment results presented in **Table 7** suggest that incorporating more data from different sensors helps to improve the overall reconstruction accuracy. However, more in-depth analysis reveals that while incorporating ETM+* (*: scan-line corrector failure) images in the input data improved the accuracy of



TM image reconstructions in 2004, it simultaneously diminished the accuracy of OLI image reconstructions in 2021. In a similar vein, the inclusion of OLI somewhat reduced the accuracy of ETM+* image reconstructions in 2021, as evidenced in **Table 8** and **Figure 16**. Despite this, these fluctuations in accuracy were relatively minor.

**Table 7: Accuracy of SDC reconstruction (of all images) with different input data settings. Metrics were averaged over the six bands, and the best results are marked in bold. (ETM+*: scan-line corrector failure)**

| Metrics | 2001 TM and ETM+ | | | 2004 TM and ETM+* | | | 2021 ETM+* and OLI | | |
|---|---|---|---|---|---|---|---|---|---|
| | TM | ETM+ | All | TM | ETM+* | All | ETM+* | OLI | All |
| RMSE | 0.0205 | 0.0227 | **0.0203** | 0.0194 | 0.0264 | **0.0187** | 0.0240 | 0.0222 | **0.0187** |
| MAE | 0.0152 | 0.0170 | **0.0150** | 0.0147 | 0.0194 | **0.0138** | 0.0182 | 0.0167 | **0.0140** |
| CC | 0.7836 | 0.7459 | **0.8056** | **0.8302** | 0.7567 | 0.8297 | 0.7894 | 0.8117 | **0.8535** |

**Table 8: Accuracy of SDC reconstruction (of images from specific sensors) with different input data settings. Metrics were averaged over the six bands, and the best results are marked in bold.**

| Metrics | 2004 TM | | 2021 ETM+ | | 2021 OLI | |
|---|---|---|---|---|---|---|
| | TM | TM and ETM+ | ETM+ | OLI and ETM+ | OLI | OLI and ETM+ |
| RMSE | 0.0213 | **0.0198** | **0.0183** | 0.0186 | **0.0172** | 0.0177 |
| MAE | 0.0161 | **0.0148** | **0.0137** | 0.0142 | **0.0125** | 0.0131 |
| CC | 0.8083 | **0.8402** | 0.8365 | **0.8503** | 0.8732 | **0.8750** |



**Figure 16: Comparison of the SDC reconstruction accuracy with different input data settings. The symbol "+" (increase) and "-"**
**(decrease) indicate the change in accuracy with the additional input data.**

## 4.6 The effectiveness of Landsat cloud masking

To mitigate cloud mask omission errors in Fmask results, the SDC generation incorporated an enhanced cloud masking approach involving a brightness-threshold filter and a time-series-outlier filter. **Figure 17** presents two cases of cloud masking results by Fmask and the enhanced cloud masking method. These cases reveal the presence of cloud or heavy aerosol pixels that remain undetected by the Fmask algorithm. Meanwhile, the enhanced cloud masking method adeptly identified and filtered out the majority of these previously undetected cloud pixels, demonstrating its effectiveness in ensuring the quality of the input Landsat data for SDC generation.

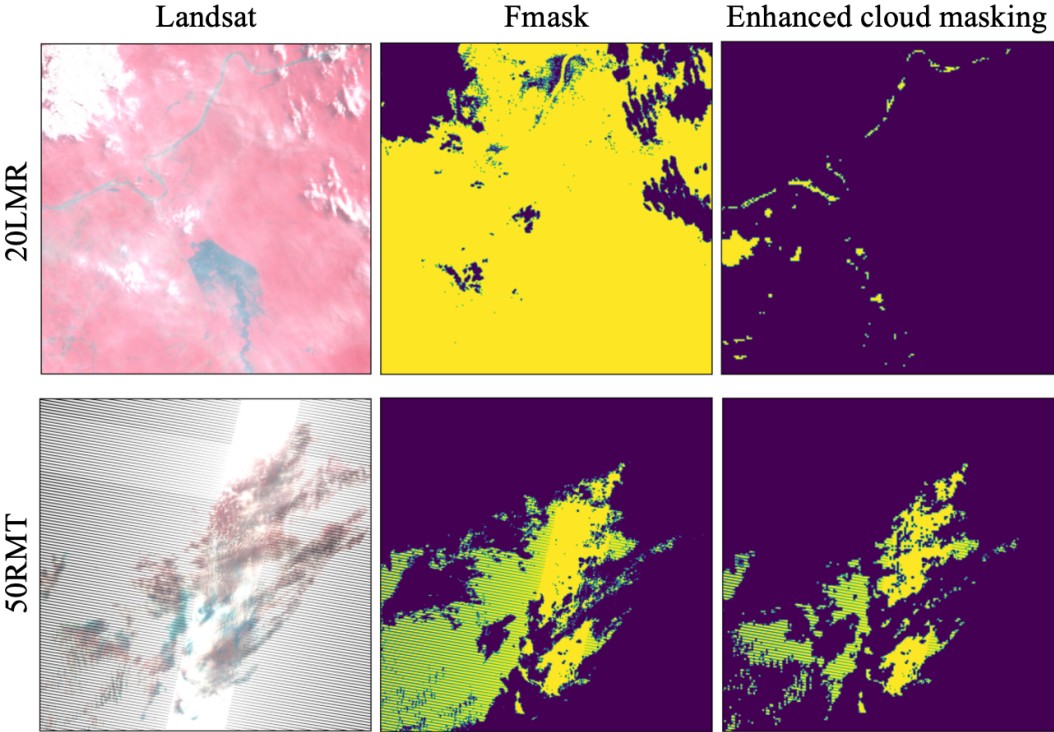

**Figure 17: Comparison of cloud masks derived by Fmask and the enhanced cloud masking approach. The yellow shading in the cloud masks signifies cloud-free pixels. It can be seen that thin clouds undetected by Fmask were mostly screened out by the enhanced cloud masking.**



## 4.7 Temporal continuity of reconstructed SDC time series

The SDC product provides an extensive historical dataset of multi-spectral and medium resolution data covering the period from 2000 to 2022. As indicated in **Table 2**, this prolonged temporal coverage encapsulates multiple distinct phases, each characterized by varied input data configurations. The temporal continuity and consistency of the SDC data emerge as pivotal factors for its efficacy in monitoring long-term land dynamics. **Figure 18** presents two distinct scenarios of SDC time series extending from 2000 to 2022. The first scenario pertains to a forest situated in a mountainous region, characterized by strong phenological changes and rugged terrain features. The second scenario depicts a desert location exhibiting minimal temporal variations and near Lambertian surfaces. It can be seen that there are no discernible discontinuities between the different phases in the SDC time series.

**Figure 18: Two cases of SDC and Landsat time series: (a) located at 35.4549° N, 113.4381° E; (b) located at 39.0850° N, 81.2460° E.**





## 5 Discussion

### 5.1 Global-scale land cover classification utilizing the SDC: a comparative analysis with Landsat composite and
530 **interpolated datasets**

Image compositing is a conventional approach employed to mitigate data gaps in optical remote sensing, which selects the highest-quality observations within a defined time interval, based on specific criteria, to create seamless clear images (Qiu et al., 2023). Contrary to best-pixel composite methods, there are also some interpolation methods to generate synthetic images based on harmonic time series fits (Zhou et al., 2022; Zhu et al., 2015). **Figure 19** displays the comparison of SDC images

with Landsat composite and interpolated images. The Landsat composite images are generated by the NLCD (Jin et al., 2023) and MAX-RNB (Qiu et al., 2023) algorithms, and the Landsat interpolated images are generated using the HAPO algorithm (Zhou et al., 2022). Notwithstanding the good visual quality apparent in these Landsat composite and interpolated images, it is evident that certain temporal change information was lost during the process of image compositing and interpolation.

Moreover, we conducted a comparative analysis to evaluate the effectiveness of using Landsat composite and interpolated images versus using SDC data for land cover classification. To perform this assessment, we utilized the validation sample set provided in previous study (Li et al., 2017), which consists a total of 32,946 data points distributed globally. To facilitate a clear and equitable comparison, we employed solely the six-band surface reflectance time series from Landsat composite, interpolated images and SDC data as input features for the classification. Subsequently, we

conducted a five-fold cross-validation procedure independently for each input data configuration, using the same LGBM classifier with default parameter settings.

**Table 9** presents the results of overall classification accuracy for various input data configurations. It is discernible that there exist minor discrepancies in performance between the NLCD and MAX-RNB results. And the utilization of seasonal composite images led to a significantly higher level of classification accuracy compared to annual composite images.

Remarkably, the highest classification accuracy was attained using SDC time series as input features, outperforming other input settings by a wide margin (2.4%~11.3%).



**Figure 19: False-color composites of Landsat, Landsat composite, and SDC images at 35.3104º N, 97.7974º W.**





**Table 9: Overall Accuracy (OA) of land cover classification results using NLCD composite images, MAX-RNB composite images, HAPO interpolated images, and SDC time series as input features.**

| Input data | NLCD | | MAX-RNB | | HAPO | SDC |
|---|---|---|---|---|---|---|
| | Seasonal | Annual | Seasonal | Annual | Daily | Daily |
| OA | 0.7179 | 0.6309 | 0.7131 | 0.6360 | 0.7197 | 0.7437 |

## 5.2 The influence of Landsat 7 ETM+ scan-line corrector failure

The scan-line corrector (SLC) of the ETM+ sensor onboard Landsat 7 failed permanently in May 2003, resulting in about 22% of the pixels per scene not being scanned (caused wedge-shaped gaps) since that time (Chen et al., 2011). These wedge-shaped gaps are evident in the Landsat ETM+ images as displayed in **Figures 17, 18, and 19**. However, this issue poses no obstacle to the SDC generation, as the uROBOT model can exploit time-series input data to fill these gaps and fuse them with MODIS in a unified manner. In the year 2012 when there are only Landsat ETM+ SLC-off images available, the uROBOT model can still utilize these ETM+ image patches as input and achieve a relatively high reconstruction accuracy, as indicated in **Table 4**. Even in later years when Landsat OLI images become available, the incorporation of Landsat ETM+ images as input can still help improve the overall accuracy of the SDC generation, as indicated in **Table 7**.

## 5.3 Limitations of SDC products

Our enhanced cloud masking approach identifies most of the previously undetected clouds. Nonetheless, there are still certain thin aerosols and cloud shadows in Landsat imagery that may undermine the data quality of the SDC dataset. Implementing a more aggressive cloud masking strategy could minimize these impacts, yet it could also significantly reduce the number of available observations. Therefore, the development of a more accurate and robust algorithm for cloud and cloud shadow detection is essential for future improvements.

As shown in **Table 8** and **Figure 16**, remaining Landsat cross-sensor inconsistencies may reduce the reconstruction accuracy slightly. And the sensor differences between Landsat and MODIS sensors could also introduce errors. Future improvements may require a more effective method for cross-sensor calibration and data harmonization.

The influence of geographic registration errors was not considered in this study, since the Landsat Collection 2 products have significantly improved the absolute geolocation accuracy of Landsat data (Crawford, 2023). However, the co-registration accuracy between Landsat imagery and MODIS data could also influence the reconstruction accuracy of SDC dataset. Future improvements could introduce co-registration processes to address this concern.





The SDC dataset could also benefit from integrating more data sources. For example, the European Space Agency (ESA) launched two satellites of the Sentinel-2 mission (S-2A and S-2B) in 2015 and 2017, respectively. The MultiSpectral Instrument (MSI) onboard both satellites acquire multi-spectral data at a spatial resolution of 10 to 60 meters depending on the wavelength with a 5-day revisit period at the equator. The incorporation of Sentinel-2 could facilitate the generation of higher-quality SDC datasets with finer spatial resolutions.

## 6 Conclusion

In this study, a global-coverage, 30-m resolution, 23-year (2000-2022), and daily-frequency Seamless Data Cube (SDC) of land surface reflectance was developed, based on the fusion of multi-sensor observations from the Landsat-5, 7, 8, 9 and MODIS Terra constellations. The SDC generation relies on a novel processing framework, which comprises a set of processing stages: gridding and reprojection, Landsat cloud masking, Landsat cross-sensor calibration, MODIS harmonization to Landsat bandpass, and a unified gap filling and spatiotemporal fusion stage. The quality of the generated SDC dataset was evaluated using a leave-one-out approach and a cross-comparison with NASA's Harmonized Landsat and Sentinel-2 products. The leave-one-out validation at 425 global test sites assessed the agreement between the SDC with actual Landsat surface reflectance values (not used as input), revealing an overall Mean Absolute Error (MAE) of 0.014 (the valid range of surface reflectance values is 0-1). The cross-comparison of the SDC with HLS products at 22 Military Grid Reference System (MGRS) tiles revealed an overall Mean Absolute Deviation (MAD) of 0.017 with L30 (Landsat-8-based 30-m HLS product) and a MAD of 0.021 with S30 (Sentinel-2-based 30-m HLS product).

The SDC has several advantages compared with other existing Landsat-based surface reflectance datasets: (i) it exhibits a higher observation frequency and enhanced capabilities for monitoring land cover changes; (ii) it is consistent in both spatial and temporal dimensions, without missing values; (iii) it has a global coverage and a prolonged 23-year duration from 2000 to 2022; (iv) validation results revealed a high level of accuracy in SDC reconstruction. Moreover, the experiment employing the SDC for global-scale land cover classification underscore its advantages comparing with Landsat composite/interpolated datasets, achieving a sizable improvement in overall accuracy (2.4%~11.3%). The SDC will be a competitive analysis-ready surface reflectance dataset in many other studies related to global environmental monitoring.

## 7 Code and data availability

The SDC dataset is available at https://doi.org/10.12436/SDC30.26.20240506 (Chen et al., 2024) or on the project website http://sdc.iearth.cloud/, where a web-based interface is provided for all researchers to freely access the SDC dataset. Notably,

the SDC dataset is dynamically generated upon request to optimize data storage efficiency. Furthermore, leveraging the generated SDC dataset, we have produced global 23-year (2000-2022) annual land cover maps using the FROM_GLC classification system, which are readily accessible on the website. All code utilized in our analyses and the accompanying

experimental data are available upon reasonable request.

## 8 Author contributions

P.G. conceptualized the research idea and supervised the project. S.C. and J.W. developed the methodologies, performed data analysis, and drafted the manuscript. Q.L and X.L. processed the MODIS data. S.C., R.L., P.Q. curated the data. J.W. and J.Y. configured and maintained the computing resources. J.B.W. developed the web-based user interface. S.C., J.W.,

S.Y., H.H., and P.G. reviewed and edited the manuscript.

## 9 Acknowledgements

This research was jointly supported by the Major Program of the National Natural Science Foundation of China (42090015), the National Natural Science Foundation of China (42071400), and the Croucher Foundation (CAS22902/CAS22HU01). We would like to thank NASA and USGS for providing the Landsat and MODIS products used in this study.


**Competing interests**. The authors declare that they have no known competing financial interests or personal relationships that could have influenced the work reported in this study.

**Disclaimer**. Publisher's note: Copernicus Publications remains neutral with regard to jurisdictional claims in published maps

and institutional affiliations.




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
