# Peer review of "Global 30-m seamless data cube (2000-2022) of land surface reflectance generated from Landsat-5,7,8,9 and MODIS Terra constellations"

_Earth System Science Data, 2024_

## Community Comment (CC9)

 **Chrono**
[Figure]

[Figure]

[Figure]

[Figure]

[Figure]

[Figure]

[Figure]

[Figure]

[Figure]

**All Tasks**

Recent

Downloading

Finished

    Image

    Video

    Audio
* * *
**CSDC30_45RWQ_2000-07.tif**

[Figure]
 blob:https://data-starcloud.pcl.ac.cn/bee78cbd-f1a6-4491-b23e-4db7867179c3

142.68 MB
* * *
**CSDC30_45RWQ_2000-06 (1).tif**

[Figure]
 blob:https://data-starcloud.pcl.ac.cn/922ef290-c04a-4e28-83b3-815e6cdea625

139.27 MB
* * *
**CSDC30_45RWQ_2000-06.tif**

[Figure]
 blob:https://data-starcloud.pcl.ac.cn/7b401a4b-b382-4740-bda4-081a885db890

139.27 MB

---

## Author Comment (AC4)

We would like to express our gratitude to the editor and reviewers for their efforts in handling and commenting on our manuscript. We highly appreciate the insightful and helpful feedback, which has significantly helped improve our manuscript. Below, we provide detailed responses along with the suggested changes to our manuscript.

**Response to Reviewer #1's comments**

**General comments**

This paper presents the development of a global 30-m seamless data cube by fusing Landsat and MODIS data, building on the authors' previous methods. This work is highly valuable for future applications requiring fine-resolution time series data. Despite the numerous algorithms developed for fusing Landsat and MODIS data or filling gaps in Landsat data, no global datasets generated by these technologies are currently available. However, the paper has several issues that need to be addressed.

We sincerely appreciate your professional review and insightful comments on our manuscript. Your feedback has been invaluable in helping us improve our article. We have made extensive revisions to our previous draft, and our detailed point-by-point responses are listed below. The line numbers mentioned below correspond to the revised manuscript with the changes highlighted.

**Specific comments**

1. Page 3, Line 70: For Landsat interpolation methods, there are techniques that do not require numerous clear-sky observations. For instance, the nearest similar pixel interpolator method only needs one clear-sky observation.

**Response 1:** Thank you for your comment. We have added the descriptions of common interpolation methods and revised the related content in our manuscript accordingly.

**Changes in manuscript:** Lines 69-75, revised the content

*"Landsat interpolation methods also provide the capability to generate seamless synthetic Landsat images (Brooks et al., 2012; Malambo and Heatwole, 2016; Yan and Roy, 2018, 2020; Zhu et al., 2015b). Linear interpolation is commonly employed to address missing values in Landsat time series (Defourny et al., 2019; Inglada et al., 2017; Tran et al., 2022), though it may not be highly effective for applications*

*like land cover classification (Che et al., 2024). To improve performance, more sophisticated interpolation methods have been developed (Brooks et al., 2012; Malambo and Heatwole, 2016; Yan and Roy, 2018; Zhou et al., 2022; Zhu et al., 2015b). Nevertheless, a significant limitation of these methods is their dependence on numerous clear-sky Landsat observations for accurate time series estimation (Chen et al., 2021; Zhu et al., 2015b)."*

2. Introduction: Before the last paragraph, the authors should discuss the current research gap. Specifically, they should explain why a full-chain processing framework and such fused data are necessary. Additionally, they should outline the challenges users face when using current methods to produce data independently.

**Response 2:** Thank you for your comment. We have added sentences to discuss the current research gap in our manuscript.

**Changes in manuscript:** Lines 96-101, added

*"To the best of our knowledge, there is currently no global, 30-m, and seamless dataset of land surface reflectance generated by fusing Landsat and MODIS products available to the community. Although there have been numerous studies dedicated to develop algorithms for missing data reconstruction and multi-sensor data fusion (Shen et al., 2015; Zhu et al., 2018), unified and generalized frameworks for effective and efficient Landsat-MODIS data fusion at global scales have not yet been explored extensively. To address this need, in this study, we ..."*

3. Page 11, Figure 3: When performing harmonization, was Landsat upscaled to the resolution of MODIS? Figure 3 suggests that both datasets have the same resolution. A linear transformation model was used; how do the authors address the issue when the linear model is not statistically significant?

**Response 3:** Thank you for your comment.

(a) We resampled MODIS data to match the 30-m spatial resolution of Landsat, as describe in Lines 189-191. This resampling operation helps streamline subsequent processing steps, and we found that its computational costs are relatively negligible.

(b) Indeed, a single linear model is insufficient for large-scale applications. Therefore, we construct multiple linear models for each local small area separately (e.g., 64×64 image patch), to better account for spatial heterogeneity. We also used an overlapping mechanism to mitigate spatial inconsistencies between neighboring image patches. The MODIS-Landsat harmonization serves to enhance the consistency between paired MODIS-Landsat images, and thereby improve the reconstruction accuracy of subsequent gap filling and spatiotemporal fusion.

**Changes in manuscript:** Lines 279-281, added

*"This approach employs multiple regional transformation models to better account for material-dependent spectral characteristics that vary across regions and uses an overlapping mechanism to enhance spatial consistency between neighboring image patches."*

4. Major Differences Between uROBOT and ROBOT Models: What are the primary differences between the uROBOT model and the previously developed ROBOT model by the authors?

**Response 4:** Thank you for your comment. There are two major differences between uROBOT and ROBOT models: (a) uROBOT has an extra penalty term $\left|F_p^+ - D_F^+\alpha\right|_2^2$, which allows it to exploit partly observed Landsat image $F_p^+$ to better reconstruct the unobserved/cloud contaminated part. The uROBOT model can do gap filling and spatiotemporal fusion in a unified manner. (b) As indicated in Eq. 8, uROBOT utilizes an interpolated Landsat image in the temporal continuity penalty term $\left|F_{interp} - D_F\alpha\right|_2^2$, while ROBOT uses the cloud-free Landsat image acquired nearest to prediction phase. This modification helps to improve the performance of uROBOT.

**Changes in manuscript:** Lines 336-338, added

*"Additionally, uROBOT utilizes the interpolated Landsat image $F_{interp}$ in the temporal continuity penalty term, which further improves the performance of uROBOT."*

5. Accuracy of the Time Series Model in Eq. 6: The accuracy of the time series model in Equation 6 could be affected by the time interval of the data. In cloudy regions, if the data is too sparse, is the result reliable? How do the authors address diverse changes when the data is sparse?

**Response 5:** Thank you for your comment. We agree that the reconstruction accuracy of SDC will decrease if the data is too sparse in cloudy regions. On one hand, we will employ multiple years of Landsat and MODIS time series images, to make sure there are sufficient input data. On the other hand, as indicated in Eq. 8, the uROBOT model has five constraints/priors:

$$\min_{\alpha}|C_p - D_C\alpha|_2^2 + \lambda|\alpha|_1 + \beta(|F_{interp} - D_F\alpha|_2^2) + \mu\left(|F_p^+ - D_F^+\alpha|_2^2\right) \qquad (8)$$

(i) $|C_p - D_C\alpha|_2^2$, $\alpha$ should be consistent with the MODIS representation;

(ii) residual distribution $\hat{F}_p = D_F\alpha + (C_p - D_C\alpha)$, the low-frequency components of $\hat{F}_p$ should be consistent with $C_p$;

(iii) $|\alpha|_1$, $\alpha$ should be sparse;

(iv) $|F_{interp} - D_F\alpha|_2^2$, $\alpha$ should be consistent with the representation of $F_{interp}$, to ensure temporal continuity;

(v) $|F_p^+ - D_F^+\alpha|_2^2$, $\alpha$ should be consistent with the representation of observed part $F_p^+$.

If the input data is still too sparse, the constraints (ii) and (iv) will function and make sure that the estimated results are consistent with $C_p$ and no worse than the interpolated images $F_{interp}$.

**Changes in manuscript:** Lines 346-348, added

*"In regions with frequent cloud cover, the scarcity of cloud-free observations can pose a challenge. To address this, the temporal continuity constraint $\beta|F_{interp} - D_F\alpha|_2^2$ and the residual distribution in **Equation (9)** ensure that our estimations are consistent with $C_p$ and are at least as accurate as the interpolated results $F_{interp}$.*

6. Eq. 7: The coefficients from MODIS are used for Landsat. This approach may be acceptable if the land is homogeneous, but it lacks a clear mechanism for complex landscapes. Landsat pixels have different temporal dependencies compared to coarse pixels. If this issue affects the reliability of the final product, the end users should be notified.

**Response 6:** Thank you for your comment. Indeed, directly transferring the coefficients from MODIS in Eq. 6 would not obtain satisfactory results. In this study, the coefficients $\alpha$ were obtained by solving the optimization problem in Eq. 8, as presented in Response 5. The five constraints in Eq. 8 will function jointly, utilizing information from both MODIS and Landsat time series images to facilitate the reconstruction of unobserved Landsat images.

Using 500-m MODIS data to reconstruct 30-m Landsat image is an under-determined problem. There are certainly extreme conditions, such as some complex landscapes and diverse changes. If the information provided in the input data is incomplete, it will be theoretically impossible to accurately reconstruct 30-m Landsat images by using 500-m MODIS data. We have revised the content in Section 5.3 to acknowledge readers about this issue.

**Changes in manuscript:** Lines 600-608, revised the content to acknowledge readers about the above-mentioned issues.

*"The SDC is not equivalent to actual daily 30-m Earth observations data. It is an estimation based on Landsat and MODIS time series observations. Reconstructing missing Landsat data is an under-determined problem, meaning there can be infinitely many possible solutions (Shen et al., 2015). By using 500-m MODIS images as "guidance" and applying the constraints presented in Equation (8), we can narrow down the solution space and make more accurate estimations. However, achieving 100% accuracy is not feasible since the information provided in the input data is usually incomplete. Additionally, the effective spatial resolution of MODIS observations changes significantly due to the variations of view angles (Pahlevan et al., 2017). Even after BRDF normalization and temporal smoothing, these effects cannot be perfectly mitigated. The effective temporal resolution of SDC depends on the quality of the input Landsat and MODIS data, which can vary in space and time."*

7. Eq. 9: The equation redistributes the residual to handle land cover changes, but the residual is at a coarse scale. How do the authors address changes at the fine pixel scale?

**Response 7:** Thank you for your comment. The regularization terms in Eq. 8 contribute to handle land cover changes at the fine pixel scale. The MODIS consistency term, the interpolated $F_{interp}$ consistency term, and the partially observed $F_p^+$ consistency term help predict step changes and gradual changes. For ephemeral changes, if there are Landsat images in the time-series input similar to the target Landsat image, the MODIS consistency term and the partially observed $F_p^+$ consistency term can help identify

these similar images and use them to predict ephemeral land cover changes. If there are no similar Landsat images available, the residual distribution in Eq. 9 will function, distributing coarse-resolution residual information as a compromised solution.

However, there are certain situations that the information provided in the input Landsat and MODIS data is incomplete for accurate reconstruction of unobserved 30-m Landsat images. We have revised the content in Section 5.3 to inform readers about this issue.

**Changes in manuscript:** Line 342, added

*"All the constraint terms in Equation (8) contribute to addressing gradual and step changes."*

Lines 600-608, added

*"The SDC is not equivalent to actual daily 30-m Earth observations data. It is an estimation based on Landsat and MODIS time series observations. Reconstructing missing Landsat data is an under-determined problem, meaning there can be infinitely many possible solutions (Shen et al., 2015). By using 500-m MODIS images as "guidance" and applying the constraints presented in Equation (8), we can narrow down the solution space and make more accurate estimations. However, achieving 100% accuracy is not feasible since the information provided in the input data is usually incomplete. Additionally, the effective spatial resolution of MODIS observations changes significantly due to the variations of view angles (Pahlevan et al., 2017). Even after BRDF normalization and temporal smoothing, these effects cannot be perfectly mitigated. The effective temporal resolution of SDC depends on the quality of the input Landsat and MODIS data, which can vary in space and time. "*

8. Page 15, Line 355: The three accuracy metrics mentioned cannot adequately assess the spatial context preserved by the fused images. Some spatial metrics, such as edge features (see examples in https://doi.org/10.1016/j.rse.2022.113002), should be presented.

**Response 8:** Thank you for your comment. We have added results calculated using Robert's edge spatial features in the accuracy validation.

**Changes in manuscript:** Lines 378-384, added

*"using standard metrics, such as Correlation Coefficient (CC), Root Mean Square Error (RMSE), Mean Absolute Error (MAE), rMAE (MAE normalized by true surface reflectance values), and Robert's edge (Edge) spatial features (Zhu et al., 2022). We calculated the normalized difference of the Edge spatial features between reconstructed image and actual Landsat image. The normalized metric values range from -1 to 1, indicating the under- or over-estimate of spatial details. The average normalized metric value of pixels with Edge value higher than 90th percentile in the actual Landsat image was used to represent the spatial accuracy of the reconstructed image (Zhu et al., 2022)."*

Revised the content in Tables 4 and 6

**Table 4: Accuracy of SDC reconstruction at the 425 global test sites (ETM+*: scan-line corrector failure).**

| Band (*mean ±std*) | 2001 (TM and ETM+) | | | | 2004 (TM and ETM+*) | | | | 2012 (ETM+*) | | | | 2021 (ETM+* and OLI) | | | |
|---|---|---|---|---|---|---|---|---|---|---|---|---|---|---|---|---|
| | RMSE | MAE | CC | Edge | RMSE | MAE | CC | Edge | RMSE | MAE | CC | Edge | RMSE | MAE | CC | Edge |
| Blue | 0.017 ±0.009 | 0.012 ±0.006 | 0.745 ±0.063 | -0.280 ±0.108 | 0.016 ±0.009 | 0.012 ±0.006 | 0.771 ±0.063 | -0.264 ±0.105 | 0.016 ±0.010 | 0.012 ±0.006 | 0.795 ±0.053 | -0.250 ±0.099 | 0.015 ±0.007 | 0.011 ±0.005 | 0.792 ±0.075 | -0.245 ±0.093 |
| Green | 0.017 ±0.008 | 0.013 ±0.005 | 0.806 ±0.053 | -0.215 ±0.098 | 0.016 ±0.008 | 0.012 ±0.006 | 0.832 ±0.055 | -0.205 ±0.096 | 0.015 ±0.009 | 0.011 ±0.006 | 0.844 ±0.050 | -0.189 ±0.087 | 0.016 ±0.008 | 0.012 ±0.005 | 0.863 ±0.058 | -0.186 ±0.080 |
| Red | 0.019 ±0.008 | 0.014 ±0.006 | 0.840 ±0.052 | -0.198 ±0.098 | 0.018 ±0.008 | 0.013 ±0.006 | 0.866 ±0.053 | -0.188 ±0.093 | 0.018 ±0.009 | 0.013 ±0.006 | 0.874 ±0.052 | -0.179 ±0.084 | 0.018 ±0.008 | 0.013 ±0.005 | 0.891 ±0.059 | -0.174 ±0.081 |
| NIR | 0.029 ±0.008 | 0.021 ±0.006 | 0.891 ±0.027 | -0.178 ±0.083 | 0.027 ±0.009 | 0.020 ±0.006 | 0.910 ±0.024 | -0.164 ±0.076 | 0.027 ±0.009 | 0.020 ±0.007 | 0.912 ±0.025 | -0.157 ±0.068 | 0.028 ±0.009 | 0.021 ±0.007 | 0.922 ±0.029 | -0.154 ±0.061 |
| SWIR1 | 0.025 ±0.003 | 0.018 ±0.002 | 0.834 ±0.022 | -0.180 ±0.092 | 0.022 ±0.004 | 0.016 ±0.003 | 0.851 ±0.037 | -0.163 ±0.082 | 0.022 ±0.004 | 0.016 ±0.003 | 0.867 ±0.037 | -0.154 ±0.071 | 0.022 ±0.003 | 0.016 ±0.003 | 0.887 ±0.033 | -0.163 ±0.068 |
| SWIR2 | 0.018 ±0.003 | 0.014 ±0.002 | 0.810 ±0.045 | -0.204 ±0.095 | 0.016 ±0.003 | 0.012 ±0.002 | 0.841 ±0.061 | -0.190 ±0.091 | 0.017 ±0.004 | 0.012 ±0.003 | 0.853 ±0.060 | -0.177 ±0.080 | 0.016 ±0.003 | 0.012 ±0.002 | 0.861 ±0.058 | -0.167 ±0.071 |

**Table 6: Cross-comparison results of between the SDC with HLS L30 and S30 products at the 22 MGRS tiles.**

| Band | Compare SDC with HLS L30 | | | | Compare SDC with HLS S30 | | | |
|---|---|---|---|---|---|---|---|---|
| | RMSD | MAD | CC | Edge | RMSD | MAD | CC | Edge |
| Blue | 0.058 | 0.014 | 0.892 | -0.217 | 0.059 | 0.017 | 0.849 | -0.339 |
| Green | 0.055 | 0.015 | 0.906 | -0.217 | 0.060 | 0.020 | 0.863 | -0.317 |
| Red | 0.056 | 0.017 | 0.920 | -0.214 | 0.060 | 0.022 | 0.890 | -0.306 |
| NIR | 0.054 | 0.023 | 0.924 | -0.218 | 0.059 | 0.027 | 0.899 | -0.233 |
| SWIR1 | 0.030 | 0.017 | 0.977 | -0.201 | 0.034 | 0.022 | 0.971 | -0.154 |
| SWIR2 | 0.025 | 0.014 | 0.980 | -0.196 | 0.028 | 0.018 | 0.974 | -0.164 |

9. Tables 4-6: Is the accuracy assessment conducted for each site? The tables only show the mean value. What is the range of these indices?

**Response 9:** Thank you for your comment. The accuracy assessment was conducted for each site, and the metric values presented in Tables 4-6 are averaged over all sites (or sites with the same dominant land cover types). We have added the corresponding standard deviation values to these tables.

**Changes in manuscript:** revised the content in Table 4, added corresponding standard deviation values, as presented in Response 8.

10. Figure 14: The RMSD values in Figure 14 are on a different scale compared to Tables 4-6.

**Response 10:** Thanks for pointing this out. We have corrected the scale of RMSD values in Figure 14.

**Changes in manuscript:** Figure 14, changed the scale of reflectance values from 0-10000 to 0-1, and re-calculated corresponding RMSD values.

|  | Blue | Green | Red | NIR | SWIR1 | SWIR2 |
|---|---|---|---|---|---|---|
| Before | 0.0087 | 0.0077 | 0.0111 | 0.0215 | 0.0187 | 0.0145 |
| After | 0.0048 | 0.0055 | 0.0082 | 0.0160 | 0.0151 | 0.0117 |
| Reduce | 44.8% | 28.5% | 26.1% | 25.7% | 18.9% | 19.4% |

**10SEH**

|  | Blue | Green | Red | NIR | SWIR1 | SWIR2 |
|---|---|---|---|---|---|---|
| Before | 0.0114 | 0.0122 | 0.0175 | 0.0388 | 0.0344 | 0.0334 |
| After | 0.0059 | 0.0080 | 0.0107 | 0.0242 | 0.0199 | 0.0194 |
| Reduce | 47.8% | 34.3% | 38.8% | 37.6% | 42.0% | 42.0% |

**17RKQ**

|  | Blue | Green | Red | NIR | SWIR1 | SWIR2 |
|---|---|---|---|---|---|---|
| Before | 0.0143 | 0.0158 | 0.0180 | 0.0495 | 0.0270 | 0.0249 |
| After | 0.0062 | 0.0066 | 0.0112 | 0.0351 | 0.0172 | 0.0157 |
| Reduce | 56.4% | 58.1% | 37.9% | 29.1% | 36.1% | 36.9% |

**32UNA**

|  | Blue | Green | Red | NIR | SWIR1 | SWIR2 |
|---|---|---|---|---|---|---|
| Before | 0.0084 | 0.0074 | 0.0098 | 0.0231 | 0.0174 | 0.0143 |
| After | 0.0044 | 0.0055 | 0.0082 | 0.0158 | 0.0150 | 0.0128 |
| Reduce | 47.6% | 25.8% | 16.3% | 31.4% | 13.6% | 9.9% |

**36MWS**

|  | Blue | Green | Red | NIR | SWIR1 | SWIR2 |
|---|---|---|---|---|---|---|
| Before | 0.0189 | 0.0165 | 0.0183 | 0.0317 | 0.0187 | 0.0195 |
| After | 0.0075 | 0.0072 | 0.0103 | 0.0260 | 0.0123 | 0.0139 |
| Reduce | 60.1% | 56.4% | 43.5% | 18.0% | 34.2% | 28.9% |

**49SGV**

|  | Blue | Green | Red | NIR | SWIR1 | SWIR2 |
|---|---|---|---|---|---|---|
| Before | 0.0097 | 0.0113 | 0.0216 | 0.0267 | 0.0407 | 0.0422 |
| After | 0.0069 | 0.0089 | 0.0164 | 0.0177 | 0.0303 | 0.0312 |
| Reduce | 29.0% | 21.4% | 24.3% | 33.8% | 25.5% | 26.0% |

**55HCC**

Figure 14: The mean RMSDs between ETM+ and OLI observations before and after the cross-calibration for six spectral bands at six MGRS tiles.

11. Table 9: It would be beneficial to include a figure showing examples where only the Seamless Data Cube (SDC) can accurately classify the pixels, whereas other data cannot.

**Response 11:** Thank you for your suggestion. We have added a figure presenting examples where the SDC can correctly distinguish data of different land cover types, while other data sources cannot.

**Changes in manuscript:** Lines 578-580, added

[revised manuscript text omitted]

---

## Author Comment (AC5)

We would like to express our gratitude to the editor and reviewers for their efforts in handling and commenting on our manuscript. We highly appreciate the insightful and helpful feedback, which has significantly helped improve our manuscript. Below, we provide detailed responses along with the suggested changes to our manuscript.

**Response to Reviewer #2's comments**

**General comments**

This submission aims at building a daily 30-m Landsat data record (2000-2022) based on Landsat 5,7,8,9 and MODIS for gap filling. Authors made sure that data are pre-processed (with AC and BRDF) and harmonized. The main pillars of the data is that they are daily at 30 m and span the 20+ years. While this dataset will certainly find a lot of applications and might be valuable to the community, and I applaud authors for taking on this challenge, in my opinion, the description of this dataset is exaggerated (e.g., daily component) and authors do not provide evidences that this is actual daily data. You cannot use 8-(16-)-day data to prove that your dataset is actual daily because you should have available daily reference data. I think this is one of the areas that authors did not work through and claim that your dataset resemble actual daily data is false. Maybe, it would have made sense to focus on regular 8-day composites. Maybe, the problem of generating daily data from 8-/16-day Landsat does not have a solution. Certainly, MODIS can help in certain regions, but I have huge doubts about its applicability globally given discrepancies in spatial resolution. Furthermore, even MODIS, majority of users use 8-day or 16-day composites given that MODIS acquires twice per day (Terra/Aqua). It's done because of clouds and because decrease in spatial resolution of the viewing angle (which increases with the cycle 1-15, and nadir only every 16 days). There are also some very dubious choices (e.g., moving the grid) . From CC comments, one can see obvious artifacts - it's understandable, from global product you can always find errors. But, as mentioned above, it's not real daily product, you did not provide evidences to claim it. I suggest authors to substantially re-work this and give a good thought on what real problem you are trying to solve and provide tangible solution that can be validated.

We sincerely thank the reviewer for the valuable feedback, which has greatly helped us to improve the quality of this manuscript. Following your suggestions, we have made extensive modifications to our manuscript to enhance the clarity of our results. The line numbers mentioned below refer to the revised manuscript with the changes highlighted.

(i) We agree that the SDC is not equivalent to actual daily 30-m EO dataset. Rather, it is an estimation based on Landsat and MODIS time series observations. By developing improved reconstruction algorithms and incorporating more useful priors/constraints, we aim to enhance the estimation accuracy as much as possible. However, achieving 100% accuracy is not feasible since the information provided in the input data is usually incomplete. The effective temporal resolution of SDC depends on the quality of input Landsat and MODIS data, which can vary in space and time. We have modified the descriptions of the daily component and revised the content in Section 5.3 to inform readers about these limitations.

(ii) Indeed, there is no daily 30-m reference dataset available. The near-daily 500-m MODIS dataset should not be used as reference data since it has been employed as input data for the generation of SDC. Therefore, we employed the leave-one-out validation method, which has been widely used to evaluate the performance of MODIS-Landsat fusion algorithms. The calculated metric values from the leave-one-out validation can serve as an indicator of the data quality of reconstructed datasets. Moreover, we also cross-compared SDC with the HLS products, which provide 30-m observations with a 2-3-day revisit frequency.

(iii) This study aims to develop a global, 30-m, seamless dataset of surface reflectance by combining Landsat and MODIS products. We attempt to find an effective approach that can preserve the temporal information and even enhance the effective temporal resolution by incorporating MODIS data. There are certainly estimation errors, and the SDC is not equivalent to actual daily 30-m EO data. Despite this, incorporating near-daily 500-m MODIS data can indeed enhance the monitoring of rapid land cover changes, such as crop harvests. We presented examples showing that using SDC reduces the time gap for confirming land cover changes, compared to using Landsat data alone. We agree that in some situations the information provided by coarse-resolution MODIS data will not be helpful for Landsat reconstruction. Therefore, we applied multiple constraints in our model described in **Equations 8-9** to ensure that our estimations are at least no worse than the interpolated Landsat images.

(iv) The issues presented in the CC comments are caused by undetected residual clouds. The referred MGRS tile 20MQB is located in Brazil, a cloudy region, with input data from the year 2000. The atmospheric correction algorithm LEDAPS for Landsat TM/ETM+ does not perform as well as the LaSRC for Landsat OLI (Vermote et al., 2018). Additionally, there are more cloud omission errors in Landsat TM/ETM+ observations due to the lack of a cirrus band (Zhu et al., 2015a). We have revised the content in Section 5.3 to inform users about the impacts of atmospheric correction and cloud detection algorithms. The data quality of the SDC dataset is significantly better in most other regions and years.

**Specific comments**

1. "It is noteworthy that our adopted grid slightly deviates from the Sentinel-2 grid. Since the original Landsat coordinate system exhibits a half-pixel (15 meters) offset relative to the Sentinel-2 grid, we expanded and shifted the original MGRS grid by 15 meters in each direction to align with the Landsat coordinate system."
First, there is no shift; it's what is used for referencing a pixel value: center in Landsat and UL in S2 (in grid) (also what is used by default in GDAL). It was a poor decision to shift the grid. Whereas, if you selected MGRS as a coordinate grid, you should have re-projected Landsat (like HLS) into MGRS. Such shift can cause artifacts when comparing to the HLS data.

**Response 1:** Thank you for pointing this out. We have revised the related content in this paragraph to clarify the differences. The USGS aligns the UTM coordinate origin with a pixel center, while the ESA aligns it with a pixel corner. The HLS products use the Sentinel-2 grid directly, requiring Landsat data to be resampled using cubic convolution to re-align Landsat pixels (HLS product documentation, *https://lpdaac.usgs.gov/documents/1698/HLS_User_Guide_V2.pdf*). We expanded the Sentinel-2 tiles by 15 meters in each direction to align them with the original Landsat coordinate system, minimizing the need for resampling Landsat data. For most Landsat images, only image cropping is needed to load them into our grid, since they are in the same UTM zone and both reference the UTM coordinate origin with a pixel center.

**Changes in manuscript:** Lines 176-181, changed from "*Since the original Landsat coordinate system exhibits a half-pixel (15 meters) offset relative to the Sentinel-2 grid, we expanded and shifted the original MGRS grid by 15 meters in each direction to align with the Landsat coordinate system.*"

to

"*The original Landsat coordinate system aligns the UTM coordinate origin with a pixel center, while the Sentinel-2 grid aligns it with a pixel corner (Claverie et al., 2018). We expanded the Sentinel-2 tiles by 15 meters in each direction to align them with the original Landsat coordinate system, minimizing the need for resampling Landsat data.*"

2. "Therefore, our approach aims at building multiple transformation models for each MGRS tile and each spectral band separately."
Building models per MGRS tile might introduce issues re spatial consistency. Did you check the impact

of such an approach on the overlapping areas (between tiles)? How are consistent those temporally to reflect land cover changes?

**Response 2:** Thank you for your comment. We used time series overlap of ETM+ and OLI observations from neighboring MGRS tiles to build linear transformation models. This approach helps improve stability and mitigate the issue of spatial inconsistency between neighboring tiles. In some MGRS tiles where there is no ETM+ and OLI overlap, we can only rely on data from neighboring tiles. The calibration and subsequent processing steps for each tile are independent, including the overlapping areas. Therefore, it would not introduce temporal variations into the reflectance time series in these overlapping areas.

**Changes in manuscript:** Lines 255-256, added "*This step helps improve stability and mitigate the issue of spatial inconsistency between neighboring MGRS tiles.*"

3. I'm very skeptical about the applicability of MODIS gap-filling for Landsat on global scale. First, almost all existing approaches, including yours, does account for changes in spatial resolution with different angles. In reality, 500-m pixel can actually decrease up to 2 km one - see
https://doi.org/10.1109/TGRS.2016.2604214
So, your assumption " the basic assumption of uROBOT is that the MODIS image $C$ can be accurately approximated by a linear combination of other similar MODIS images in the input timeseries data" is only valid under the condition that spatial resolution is invariant. That's not the case, especially in cloud-prone regions. Another issue that will not work in areas of small ag fields, e.g., Arica, SE Asia, etc.
I have seen multiple times examples when a bare ground field between two vegetative fields will be brightened in MODIS when resolution decreases. And used MODIS-based vegetation signal for Landsat will introduce huge errors. Therefore, a study must be conducted to explore how spatial resolution impacts restoration.

**Response 3:** Thank you for your comment. Indeed, the effective spatial resolution of MODIS observations can vary significantly over time. Even after BRDF normalization and temporal smoothing, these effects cannot be perfectly mitigated. Therefore, we are not directly using MODIS data to fill the data gaps in Landsat images in this study. Instead, we have adopted two constraints (i) and (ii) in the optimization problem below, utilizing MODIS information to facilitate the reconstruction of unobserved Landsat images.

$$\min_{\alpha}|C_p - D_C\alpha|_2^2 + \lambda|\alpha|_1 + \beta(|F_{interp} - D_F\alpha|_2^2) + \mu\left(|F_p^+ - D_F^+\alpha|_2^2\right) \qquad (8)$$

(i) $|C_p - D_C\alpha|_2^2$: the coefficient vector $\alpha$ should be consistent with the MODIS representation;

(ii) residual distribution $\hat{F}_p = D_F\alpha + (C_p - D_C\alpha)$: the low-frequency components of $\hat{F}_p$ should be consistent with $C_p$;

(iii) $|\alpha|_1$: $\alpha$ should be sparse;

(iv) $|F_{interp} - D_F\alpha|_2^2$: $\alpha$ should be consistent with the representation of $F_{interp}$, to ensure temporal continuity;

(v) $|F_p^+ - D_F^+\alpha|_2^2$: $\alpha$ should be consistent with the representation of observed part $F_p^+$.

Reconstructing missing Landsat data is an under-determined problem, meaning there can be infinitely many possible solutions. By using 500-m MODIS images as "guidance", we can narrow down the solution space and make more accurate estimations. However, the information provided by 500-m MODIS data is usually incomplete for perfect reconstruction of 30-m Landsat images. Therefore, more useful constraints/priors, such as those in **Equation 8**, are needed to ensure that the obtained results are as accurate as possible. We agree that in some situations the information provided by coarse-resolution MODIS data will not be helpful for Landsat reconstruction. Therefore, we applied multiple constraints in our model described in **Equations 8-9** to ensure that our estimations are at least no worse than the interpolated Landsat images.

We have conducted experiments to study how the spatial resolution of MODIS input data impacts SDC reconstruction accuracy. The conclusion is intuitive: the usable information reduces as the spatial resolution of input MODIS data decreases, leading to higher level of uncertainty and prediction errors. We agree that the view angle information of MODIS data used for SDC restoration should be added to the QA band to inform users about the potential variations of restoration accuracy.

**Changes in manuscript:** Lines 605-608, added
*"Additionally, the effective spatial resolution of MODIS observations changes significantly due to the variations of view angles (Pahlevan et al., 2017). Even after BRDF normalization and temporal smoothing, these effects cannot be perfectly mitigated. The effective temporal resolution of SDC depends on the quality of the input Landsat and MODIS data, which can vary in space and time."*
Lines 346-348, added

*"In regions with frequent cloud cover, the scarcity of cloud-free observations can pose a challenge. To address this, the temporal continuity constraint $\beta \left| F_{interp} - D_F \alpha \right|_2^2$ and the residual distribution in* **Equation (9)** *ensure that our estimations are consistent with $C_p$ and are at least as accurate as the interpolated results $F_{interp}$."*

4. Section 5.3 must be re-written as it does not show limitations but rather what have not been done. There should be paragraphs re snow, re coastal regions and water, as AC algorithms do not work the best there (especially for snow as retrieval of aerosols is extremely difficult there). Furthermore, probably this product is not applicable to detecting rapid changes (daily) in land cover such as constructions as it will depend on actual acquisitions (and not blended daily). It will probably will not allow to detect "daily" burned fields or harvested fields because the signal will change in 2-3 days, or landslides, or iceberg movement, or fire propagation - anything that truly changes every hour or day. Again, in reality your daily product will not allow (please, prove me wrong!) detection of these events (at daily basis) which require true daily data.

**Response4:** Thank you for your comment. We have revised the content in Section 5.3 to acknowledge that the effective temporal resolution of SDC can vary depending on the input data conditions. We also informed readers that atmospheric correction algorithms may not work optimally in areas with snow and water bodies. The LEDAPS atmospheric correction algorithm for Landsat-5,7 does not perform as well as the LaSRC for Landsat-8,9 (Vermote et al., 2018). Additionally, the Fmask algorithm for Landsat-5,7 is less effective compared to its performance for Landsat-8,9, due to the absence of the cirrus band (Zhu et al., 2015a). Residual aerosols and omitted clouds in Landsat input data can introduce temporal noise and spatial artefacts in SDC data.

We agree that the SDC is not equivalent to actual daily 30-m EO data. It is an estimation based on Landsat and MODIS time series observations. By developing improved reconstruction algorithms and incorporating more useful priors/constraints, we aim to enhance the estimation accuracy as much as possible. However, achieving 100% accuracy is not feasible since the information provided in the input data is usually incomplete. The effective temporal resolution of SDC depends on the quality of the input Landsat and MODIS data, which can vary in space and time.

*30-m 8/16-day Landsat < SDC < actual daily 30-m EO data*

*(we aim to minimize the gaps between the SDC and actual daily 30-m EO data as much as possible)*

Despite these limitations, SDC has notable strengths. As demonstrated in the figure below, incorporating near-daily 500-m MODIS data can indeed enhance the monitoring of some rapid land cover changes, such as crop harvests. Although there are differences between reconstructed SDC data and actual HLS S30 observations, using SDC reduces the time gap for confirming land cover changes, compared to using Landsat data alone. Moreover, SDC is consistent in both spatial and temporal dimensions, making it analysis-ready for subsequent applications.

[Figure]

**Figure.** An example from the manuscript. Blue circles highlight the differences between reconstructed SDC and actual Sentinel-2 observations.

**Changes in manuscript:** Lines 600-611, revised the content in Section 5.3, to inform readers about the above-mentioned limitations of SDC,

*"The SDC is not equivalent to actual daily 30-m Earth observations data. It is an estimation based on Landsat and MODIS time series observations. Reconstructing missing Landsat data is an under-determined problem, meaning there can be infinitely many possible solutions (Shen et al., 2015). By using 500-m MODIS images as "guidance" and applying the constraints presented in Equation (8), we can narrow down the solution space and make more accurate estimations. However, achieving 100% accuracy is not feasible since the information provided in the input data is usually incomplete. Additionally, the effective spatial resolution of MODIS observations changes significantly due to the variations of view angles (Pahlevan et al., 2017). Even after BRDF normalization and temporal*

*smoothing, these effects cannot be perfectly mitigated. The effective temporal resolution of SDC depends on the quality of the input Landsat and MODIS data, which can vary in space and time.*

*The LEDAPS atmospheric correction algorithm for Landsat TM and ETM+ does not perform as well as the LaSRC for Landsat OLI (Vermote et al., 2018). Additionally, the Fmask algorithm for Landsat-5,7 is less effective compared to its performance for Landsat-8,9, due to the absence of the cirrus band (Zhu et al., 2015a)."*

**References**

Claverie, M., Ju, J., Masek, J.G., Dungan, J.L., Vermote, E.F., Roger, J.-C., Skakun, S.V., Justice, C., 2018. The Harmonized Landsat and Sentinel-2 surface reflectance data set. Remote Sensing of Environment 219, 145–161. https://doi.org/10.1016/j.rse.2018.09.002

Pahlevan, N., Sarkar, S., Devadiga, S., Wolfe, R.E., Roman, M., Vermote, E., Lin, G., Xiong, X., 2017. Impact of Spatial Sampling on Continuity of MODIS–VIIRS Land Surface Reflectance Products: A Simulation Approach. IEEE Trans. Geosci. Remote Sensing 55, 183–196. https://doi.org/10.1109/TGRS.2016.2604214

Shen, H., Li, X., Cheng, Q., Zeng, C., Yang, G., Li, H., Zhang, L., 2015. Missing Information Reconstruction of Remote Sensing Data: A Technical Review. IEEE Geoscience and Remote Sensing Magazine 3, 61–85. https://doi.org/10.1109/MGRS.2015.2441912

Vermote, E., Roger, J.C., Franch, B., Skakun, S., 2018. LaSRC (Land Surface Reflectance Code): Overview, application and validation using MODIS, VIIRS, LANDSAT and Sentinel 2 data's, in: IGARSS 2018 - 2018 IEEE International Geoscience and Remote Sensing Symposium. Presented at the IGARSS 2018 - 2018 IEEE International Geoscience and Remote Sensing Symposium, IEEE, Valencia, pp. 8173–8176. https://doi.org/10.1109/IGARSS.2018.8517622

Zhu, Z., Wang, S., Woodcock, C.E., 2015a. Improvement and expansion of the Fmask algorithm: cloud, cloud shadow, and snow detection for Landsats 4–7, 8, and Sentinel 2 images. Remote Sensing of Environment 159, 269–277. https://doi.org/10.1016/j.rse.2014.12.014